# FairNVT: Fair Classification via Noise Injection in Vision Transformers

## Abstract

This paper presents **FairNVT**, a lightweight debiasing framework for pretrained transformer-based encoders that improves prediction fairness while preserving task performance. FairNVT is motivated by the intuition that reducing sensitive-attribute information in the representation used by the downstream classifier can facilitate fairer predictions. Our approach learns task-relevant and sensitive embeddings via lightweight adapters, applies calibrated Gaussian noise to the sensitive embedding, and fuses it with the task representation. Together with orthogonality constraints and fairness regularization, these components jointly reduce sensitive-attribute leakage in the learned embeddings and encourage fairer downstream predictions. Across three datasets spanning vision and language, FairNVT reduces sensitive-attribute attacker accuracy, improves fairness metrics such as demographic parity difference and equalized odds, and maintains competitive task performance.

## 1 Introduction

Modern machine learning systems largely follow a pretrain–transfer paradigm, where large foundation models are trained on massive, noisy datasets and then adapted to downstream tasks using their transferable representations. Yet these models often encode *social and demographic biases* present in their training data, leading to systematic unfairness across attributes such as gender, race, and age. When deployed in sensitive domains, such as recruitment, credit scoring, or facial recognition, these biases may propagate to downstream tasks, resulting in inequitable treatment of individuals and undermine the reliability of deploying machine learning systems (Gallegos et al., 2024; Li et al., 2024).

Fairness in machine learning is commonly studied at the *prediction level*, where the goal is to ensure that model outputs satisfy group-based criteria such as demographic parity, equal opportunity, or equalized odds. These metrics quantify disparities in model predictions across demographic groups and are essential for evaluating the societal impact of deployed models. In a typical downstream pipeline, a classifier is trained on top of pretrained embeddings while satisfying these fairness criteria. However, prediction-level fairness provides only a partial view of the problem. Even when a downstream classifier satisfies fairness metrics, the underlying embeddings may still retain substantial information about sensitive attributes. Such information can often be recovered from the learned representations, enabling downstream misuse, model inversion, or fairness degradation when the embeddings are reused for new tasks (Feng et al., 2023; Gallegos et al., 2024). These observations motivate *representation-level* fairness, which aims to learn embeddings that remain predictive for downstream tasks while being invariant, or at least less informative, with respect to sensitive attributes.

A growing body of work has explored these two notions of fairness, but they are typically studied in isolation. Prediction-level methods directly impose fairness constraints on model outputs (Kang et al., 2022; Wang et al., 2023; Xie et al., 2024), whereas representation-level methods seek to reduce sensitive information in learned embeddings using adversarial learning (Zhang et al., 2018; Götte, 2023), contrastive learning (Park et al., 2022), or projection-based techniques (Islam et al., 2024; Shi et al., 2024). Recent studies further suggest that representation-level and prediction-level fairness are not necessarily aligned, highlighting the challenge of improving both simultaneously (Shen et al., 2022).

In this work, we show that reducing sensitive information in learned representations through targeted noise injection can facilitate fairer downstream predictions. We propose **FairNVT**, a lightweight debiasing framework for pretrained transformer-based encoders built on noise injection. Starting from a frozen pretrained encoder, FairNVT learns separate task and sensitive adapters, where an orthogonality objective encourages the two branches to capture complementary information. We apply calibrated Gaussian noise to the sensitive branch to reduce its utility for downstream prediction, and apply fairness constraints to mitigate residual sensitive information that remains in the task representation. Empirically, FairNVT consistently reduces sensitive-attribute predictability from the learned embeddings, improves prediction-level fairness, and maintains competitive task performance across vision and language datasets, yielding a favorable fairness–utility trade-off.

In summary, this paper makes the following three key contributions:

- We propose FairNVT, a lightweight, single-stage debiasing framework for pretrained transformer encoders that learns separate task and sensitive branches, suppresses the sensitive branch through calibrated Gaussian noise, and combines it with fairness optimization constraint to improve downstream fairness.

- We demonstrate empirically that reducing sensitive-attribute leakage in the representation used by the downstream classifier can facilitate fairer predictions while preserving task performance.

- Extensive experiments on vision and language benchmarks show that FairNVT consistently reduces sensitive-attribute predictability, improves prediction-level fairness, and maintains competitive task performance, yielding a favorable fairness–utility trade-off.

## 2 Preliminaries: Fairness in Classification

We consider two complementary notions of fairness: *prediction-level* fairness and *representation-level* fairness. *Prediction-level* fairness is commonly studied under group fairness criteria, which seek to reduce the dependence between model predictions and sensitive attributes. For example, for data-label pair $(X, Y)$, where the features $X$ may encode sensitive information $S$, a classifier $f : X \rightarrow \hat{Y}$ is considered fair with respect to demographic parity (DP) difference if $P(\hat{Y} \mid S = s) = P(\hat{Y} \mid S = s')$, i.e., the prediction distribution is invariant to the sensitive attribute. Such notion of fairness is widely studied in prior work, including Park et al. (2022); Tian et al. (2024); Park & Byun (2024).

*Representation-level* fairness instead considers the learned representations. A common objective is to learn representations that are statistically independent of sensitive attributes, i.e. $Z \perp\!\!\!\perp S$. Since exact independence is difficult to verify empirically Liu et al. (2022), prior work typically evaluates representation-level fairness by measuring how accurately the sensitive attribute can be inferred from the learned representation (Ravfogel et al., 2020; Kumar et al., 2023).

These two notions are closely related: when predictions depend only on the learned representation $\hat{Y} = h(Z)$, statistical independence $Z \perp\!\!\!\perp S$ implies $P(\hat{Y} \mid S = s) = P(\hat{Y} \mid S = s') = P(\hat{Y})$. Although exact independence is generally unattainable in practice, this observation motivates our approach, which reducing sensitive-attribute information in the representation provided to the downstream classifier can facilitate fairer predictions. We discuss the related works in Appendix A, and further discuss this motivation in Appendix B. Next, we outline how our model supports these objectives.

## 3 FairNVT Framework

FairNVT mitigates bias in downstream predictions from frozen pretrained embeddings by encouraging the separation of task-relevant and sensitive-attribute information, selectively suppressing the sensitive branch, and jointly optimizing for fairness. Figure 1 provides an overview of the proposed framework. The key design principle is to reduce the sensitive-attribute information available to the downstream task classifier while preserving task-relevant features. To achieve this goal, FairNVT consists of three key components. First, the

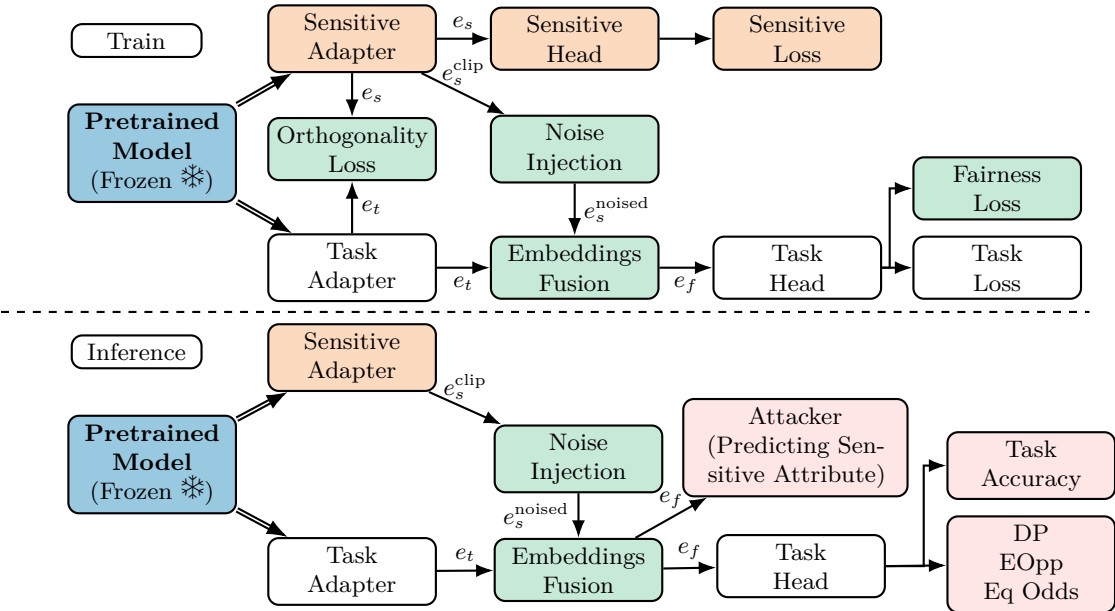

Figure 1: **Overview of the proposed FairNVT framework.** During *training*, a frozen ViT backbone is attached with lightweight task and sensitive adapters, where trainable parameters are attached to each Transformer layer (indicated by ⇒). The adapters yield task ($e_t$) and sensitive ($e_s$) embeddings. The sensitive path inputs $e_s$ for the sensitive head and a clipped and noised embedding $e_s^{\text{noised}}$ ($e_s^{\text{clip}}$ injected with noise), that is concatenated with $e_t$ to get the fused embedding $e_f$ for task prediction. We jointly optimize a weighted sum of task and sensitive classification losses, orthogonality and fairness losses. During *inference*, the sensitive and task adapters remain frozen at trained weights. A random noise is drawn from the same distribution during training to obfuscate the sensitive embedding and fused with the task embedding before entering the task classification head. Fairness metrics are calculated from the fused embedding $e_f$. Blue marks the frozen backbone; Orange extracts sensitive information; Green performs debiasing, fusion, and prediction; Red marks the evaluation metrics.

frozen backbone representation is connected to separate Task and Sensitive Adapters, each supervised by its corresponding classification objective. This split-adapter architecture encourages the Sensitive Adapter to capture sensitive-attribute-related information in a dedicated branch while preserving task-relevant information in the Task Adapter. An orthogonality objective further encourages the two branches to learn complementary representations. Second, calibrated Gaussian noise is injected into the Sensitive Adapter before it is fused with the Task embedding, suppressing the sensitive information in the pathway while retaining task-relevant features. Third, the adapters and classification heads are jointly optimized using task and sensitive classification losses together with an orthogonality loss and a fairness loss. While the classification and orthogonality objectives encourage the two branches to learn separated information, the fairness loss acts on the downstream task classifier to mitigate the residual sensitive information that remains in the fused representation. Section 3.1 discusses the model components. Section 3.2 introduces the optimization objectives, and 3.3 describes the training and inference procedures.

## 3.1 Model Components

**Adapters and classification heads.** We use the Adapter modules to extract task-relevant and sensitive information from the frozen pre-trained models, with supervision from the task and sensitive labels. The Adapters are model-agnostic, lightweight blocks of trainable parameters attached to various blocks of the frozen pre-trained model. For example, for the image classification tasks which we use the Vision Transformer (ViT, Dosovitskiy et al. (2021)) as frozen pre-trained models, the Adapters [1] are bottleneck feed-forward layers

---

[1]The Bottleneck Adapter architecture is shown in https://docs.adapterhub.ml/methods.html#bottleneck-adapters

attached to each Transformer layer, consisting of down-projection matrix to project the hidden states into a lower dimension layer, and an up-projection matrix to project back into the original hidden dimension (Poth et al., 2023). To encourage the model to learn separate task and sensitive representations, we attach separate adapters together with their corresponding classification heads. We use the class token representation (`[CLS]` token embedding) from the adapted ViT model as the task ($e_t$) and sensitive ($e_s$) embeddings, with only the task and sensitive Adapters activated respectively. Each embedding is passed to a lightweight multi-layer perceptron (MLP) that predicts its corresponding label. As described in Section 3.2, the orthogonality objective further encourages these two embeddings to capture complementary information, providing a dedicated branch on which subsequent noise injection can be applied.

**Noise injection.** To introduce perturbation to the targeted sensitive information, we clip the sensitive embedding and add random noise sampled from a Gaussian distribution. Specifically, let $e_s$ be the sensitive embedding vector, we clip the sensitive embedding to upper-bound its $L2$-norm to $C$, $e_s^{\mathrm{clip}} = e_s / \max(1, \frac{||e_s||_2}{C})$, where $C$ is a hyperparameter to control the embedding scale. Clipping bounds the scale of the embedding, allowing the subsequent noise magnitude to be calibrated relative to a fixed norm. The noise $z$ is then randomly drawn from a Gaussian distribution with mean zero and variance scales with $C$, i.e., $z \sim \mathcal{N}(0, C^2\sigma^2\mathbb{I}^d)$, where $\sigma$ controls the noise level and $d$ is the dimension of $e_s$. Finally, the perturbed embedding is obtained by adding noise $z$ isotopically to the sensitive embedding, $e_s^{\mathrm{noised}} = e_s^{\mathrm{clip}} + z$. Using zero-mean Gaussian noise obfuscates the sensitive information while avoiding a systematic shift in the embedding.

**Embedding fusion.** After perturbing the sensitive embedding, we concatenate the noised sensitive embedding with the task embedding to form the representation used by the downstream task classifier. We adopt two steps to encourage the Task and Sensitive Adapter to capture the corresponding information. First, only the clean sensitive embedding ($e_s$) is supervised by the sensitive classification head, while the noised embedding ($e_s^{\mathrm{noised}}$) is provided only to the downstream task classifier. This allows the Sensitive Adapter to continue learning sensitive-attribute-related information from clean supervision while the downstream classifier receives only the perturbed information. Second, we stop the gradient from the task classification loss from flowing through $e_s^{\mathrm{noised}}$, such that the learned task information does not interfere with the Sensitive Adapter.

## 3.2 Optimization Objectives

The proposed framework is trained jointly with classification, orthogonality and fairness losses to balance between making accurate and fair task predictions. We describe each loss and their objectives in this section.

**Classification loss.** The cross-entropy loss is used for each classification head to evaluate the predictive performances. Let $i, k$ be the sample and class index, $\theta$ be the model parameters, $(x_i, y_i)$ be each data-label pair, $\hat{y}$ be the predicted label, then the cross-entropy loss for predicting task ($t$) and sensitive ($s$) labels are,

$$L_{\mathrm{ce}}^{\alpha}(\theta) = -\frac{1}{n}\sum_{i=1}^{n}\sum_{k=1}^{K} y_{i,\alpha}^{k} \log p_{\theta}(\hat{y}_{i,\alpha} = k|x_i), \alpha \in \{s, t\}. \tag{1}$$

**Orthogonality loss.** Since the Task and Sensitive Adapters are initialized from the same frozen backbone, their learned representations may initially contain similar information. Moreover, when the task label is highly correlated with the sensitive attribute (e.g., predicting *wearing glasses* when the sensitive attribute is *age*), the task objective alone may encourage the Task Adapter to exploit sensitive-attribute information. To encourage the two branches to separate, we combine a cosine-similarity loss between the task and sensitive embeddings with the Hilbert–Schmidt Independence Criterion (HSIC) (Gretton et al., 2005) between the task embedding and the sensitive attribute labels. Let $e_t, e_s$ denote the task and sensitive embedding that depend on $\theta$ and let $s$ denote the vector of sensitive attributes,

$$L_{\mathrm{orth}}(\theta) = \mathrm{CosSim}(e_t, e_s) + \mathrm{HSIC}(e_t, s), \tag{2}$$

where $\mathrm{CosSim}(e_t, e_s) = \frac{1}{n}\sum_{i=1}^{n}(e_{t,i}^{\top}e_{s,i}/||e_{t,i}||_2||e_{s,i}||_2)^2$, and $\mathrm{HSIC}(e_t, s) = \frac{1}{(n-1)^2}\mathrm{tr}(KHLH)$ where $K_{ij} = \exp(-||e_{t,i} - e_{t,j}||_2^2/2\sigma^2)$ is the kernel matrix of the embeddings, $L_{ij} = \mathbb{1}(s_i = s_j)$ is the kernel matrix of the

sensitive attributes, and $H = \mathbf{I}_n - \frac{1}{n}\mathbf{1}_n\mathbf{1}_n^\top$ is the centering matrix. The cosine-similarity term encourages the Task and Sensitive Adapters to learn complementary representations, whereas the HSIC term reduces the statistical dependence between the task embedding and the sensitive attribute.

**Fairness loss.** While the orthogonality loss and noise injection help disentangling and suppressing the sensitive information seen by the classifier head, the remaining sensitive information flow in from the task path is handled with a fairness loss to ensure making fair predictions in the task classifier. Following the definition of demographic parity difference (Agarwal et al., 2019), let $n_0, n_1$ be the number of samples in a batch belonging to sensitive group $0, 1$ respectively, $p = p_\theta(\hat{y} = 1|x)$ be the probability of predicting the positive class of label $y$, and $\mathbf{1}[\cdot]$ be the indicator function then,

$$L_{\mathrm{dp}}(\theta) = \left| \frac{1}{n_0} \sum_{i=1}^{n_0} \mathbf{1}[s_i = 0]p_i - \frac{1}{n_1} \sum_{j=1}^{n_1} \mathbf{1}[s_j = 1]p_j \right|. \tag{3}$$

We optimize demographic parity during training because it provides a simple and stable surrogate that penalizes disparities in predictions across sensitive groups. In contrast, objectives such as equalized odds or equal opportunity additionally condition on the task label, resulting in more complex and potentially less stable optimization. Although the proposed loss explicitly optimizes only demographic parity, encouraging the classifier to rely less on sensitive-attribute information can also improve other group fairness metrics in practice. We therefore report demographic parity, equalized odds, and equal opportunity as evaluation metrics.

The overall loss is weighted to adjust the scale differences between the three losses and to allow flexibility of viewing different importance of the optimization targets, $L = L_{\mathrm{ce}}^{\mathrm{t}} + \beta_1 L_{\mathrm{ce}}^{\mathrm{s}} + \beta_2 L_{\mathrm{orth}} + \beta_3 L_{\mathrm{dp}}$, where $\beta$s are hyperparameters representing weights on each loss.

### 3.3 Training and Inference Procedures

The arrows in Figure 1 show the forward pass direction. During backpropagation, only the Adapters and classification heads parameters are updated with loss $L$, while the pre-trained model remains frozen. The noise injection and embedding concatenation steps do not induce learnable parameters. During training, lightweight task and sensitive adapters are attached to the frozen backbone to produce the task embedding $e_t$ and sensitive embedding $e_s$. The sensitive embedding is clipped and noised to obtain the embedding $e_s^{\mathrm{noised}}$, which is then concatenated with $e_t$ to form the fused embedding $e_f = [e_t, e_s^{\mathrm{noised}}]$. The fused embedding is used as the input to the task classifier head. The adapters and classifier parameters are jointly optimized with loss $L$ to encourage fairer task predictions. During inference, the adapters and classifier head are fixed at the learned parameters and applied directly to the input data.

We make the following key remarks for the training and inference pipeline. First, noise is sampled randomly from a fixed distribution and injected during both training and inference stages. Using the same perturbation mechanism at both stages ensures that the classifier is trained and evaluated under consistent conditions, encouraging the classifier to rely less on information carried by the Sensitive Adapter at inference time. Second, fairness evaluations are conducted on the fused embedding $e_f$, as it is the representation used by the trained classifier and therefore determines the model's predictions and reflects the information accessible to the decision-making process. Finally, the sensitive attribute labels are only used during training to supervise the Sensitive Adapter and compute the fairness objective. They are not accessed during inference, as the classifiers remain fixed at the trained parameters.

## 4 Experiments

We examine the performance of FairNVT on image classification tasks using the CelebA and UTKFace datasets, with a pretrained ViT-B/16 model as the frozen backbone. Across multiple task and sensitive attribute pairs, FairNVT demonstrate strong performance, achieving high task accuracy while improving prediction fairness with respect to the sensitive attribute (§4.1). We further validate our hypothesis that suppressing sensitive information through controlled noise can improve prediction-level fairness while preserving task-relevant

signals (§4.2). Implementation details are discussed in Appendix C. In appendix D.2, we extend FairNVT to the text domain and present results on the BIOS dataset, where a pre-trained Bert-Base model is used as the frozen backbone.

**Datasets[2] and tasks.** We use publicly available datasets CelebA (Liu et al., 2015) and UTKFace (Zhang et al., 2017) for facial attribute classification. CelebA contains roughly 200K images with attribute annotations. Following prior work (Tian et al., 2024; Park et al., 2022), we consider perceived gender or age as sensitive attributes, and we study target attributes such as expression(smiling), big nose and wavy hair. We use the official train/validation/test splits. UTKFace contains approximately 20K images with annotations including gender and age. To follow a binary fairness formulation (Park et al., 2022), we group age into $< 35$ vs. $\geq 35$ and use age as the sensitive attribute and gender as the target attribute [3]. The sensitive attribute of Age (Young) in CelebA is imbalanced as 77% majority vs. 23% minority, whereas the Gender (Male) attribute is more balanced as 57% majority vs. 43% minority. The Age attribute in UTKFace has 64% majority vs. 36% minority.

**Metrics.** We evaluate task performance, prediction-level and representation-level fairness performances with standard metrics[4] used by baseline methods. All reported values are scaled by $\times 10^2$.

(1) Task performance is measured using *accuracy* (Acc). We additionally report *balanced accuracy* (BAcc), as the task attribute is often imbalanced, and relying solely on accuracy may lead to misleading evaluations of task performance.

$$Acc := \frac{\mathrm{TP} + \mathrm{TN}}{\mathrm{P} + \mathrm{N}}, \; BAcc := \frac{1}{2}\left(\frac{\mathrm{TP}}{\mathrm{TP} + \mathrm{FN}} + \frac{\mathrm{TN}}{\mathrm{TN} + \mathrm{FP}}\right), \tag{4}$$

where P, N denote real positive and negatives, TP, TN, FP, and FN denote true/false positives/negatives, respectively. To assess prediction-level fairness, we employ three widely used group fairness metrics. Given true labels $Y \in \{0, 1\}$ and predictions $\hat{Y}$, let $S \in \{0, 1\}$ denote the binary sensitive attribute:

(2) *Demographic Parity (DP)* computes the difference between the largest and smallest rates across all groups:

$$DP := \max_s \mathbb{E}[\hat{Y} \mid S] - \min_s \mathbb{E}[\hat{Y} \mid S], \tag{5}$$

which simplifies to $|\mathbb{E}[\hat{Y} = 1 \mid S = 0] - \mathbb{E}[\hat{Y} = 1 \mid S = 1]|$ in the binary case.

(3) *Equalized Odds (EO)* adds conditioning on the task label compared to *DP*. EO evaluates group fairness by averaging disparities in both true positive rates and false positive rates across sensitive groups:

$$EO := \frac{1}{2}\sum_{y \in \{0,1\}} \left|\mathbb{E}[\hat{Y} = 1 \mid Y = y, S = 0] - \mathbb{E}[\hat{Y} = 1 \mid Y = y, S = 1]\right|, \tag{6}$$

(4) *Equal Opportunity (EOpp)* is a relaxed version of *EO* that only considers conditional expectations with respect to positive task labels. EOpp considers the disparities in true positive rates only:

$$EOpp := |\mathbb{E}[\hat{Y} = 1 \mid Y = 1, S = 0] - \mathbb{E}[\hat{Y} = 1 \mid Y = 1, S = 1]|. \tag{7}$$

We report absolute DP/EO/EOpp gaps throughout, in all three metrics, lower values indicate higher fairness level. To assess representation-level fairness, we examine the prediction accuracy of the sensitive attribute from an attacker. Lower values indicate higher fairness level.

(5) *Attacker accuracy/balanced accuracy (Att.Acc/BAcc)* [5] measures sensitive-information leakage using a post-hoc attacker: a MLP trained to predict $S$ from embeddings at a saved checkpoint (encoder frozen). Lower attacker accuracy indicates less recoverable sensitive information. Architecture and training details of the attacker model are provided in Appendix C.

---

[2]The datasets are publicly available and include perceived annotations provided by the dataset creators. We use these labels solely for modeling and fairness evaluation to compare with previously published results on these benchmarks.

[3]Though there is no official train/validation/test splits, the dataset has three subsets, where we use subsets 1, 2, and 3 for training, validation, and testing, respectively.

[4]https://fairlearn.org/

[5]Attacker performance is used in ablation studies only to analyze sensitive attribute leakage in the framework.

**Baselines.** We compare our approach with the vanilla setup, and several image-based fair classification baselines under a unified evaluation protocol.

- **Vanilla (ViT)** (Dosovitskiy et al., 2021): ViT with a task adapter and classification head trained, with no fairness intervention.

- **ViT-FSCL** (Park et al., 2022): Debiasing through contrastive learning; we re-implement it on a ViT backbone for consistent comparison.

- **FairViT** (Tian et al., 2024): Debiasing via adaptive masking on ViT attention maps.

- **FairVPT** (Park & Byun, 2024): Debiasing using Visual Prompt Tuning that adapts pre-trained ViT model to downstream classification tasks [6].

- **FairNet** (Zhou et al., 2025): Debiasing via a bias detector and conditional LoRA.

### 4.1 Main Results

We compare FairNVT with the baselines on CelebA and UTKFace datasets. On CelebA, we evaluate three task–sensitive attribute pairs: Expression(Smiling)/Gender(Male), Big Nose/Age(Young), and Wavy Hair/Gender(Male). On UTKFace, we evaluate the Gender/Age task–sensitive attribute pair.

**Fairness–Utility tradeoff.** Figure 2 compares the Pareto curves of task performance and fairness scores for all methods. Each point corresponds to the test performance of a model trained with a different hyperparameter configuration. As higher task accuracy (or balanced accuracy) and lower fairness metric values indicate better performance, the lower-right corner represent more favorable fairness–utility trade-offs. The points on the Pareto curve thus consist of results from configurations for which no other configuration achieves both higher task performance and better fairness simultaneously.

Overall, FairNVT consistently lies on or near the Pareto frontier, indicating a more favorable fairness–utility trade-off than competing methods. Across the four experiments, FairNVT generally achieves higher accuracy (or balanced accuracy) at the same level of fairness, or equivalently, lower demographic parity (DP), equalized odds (EO), and equal opportunity (EOpp) violations for comparable predictive performance. For the Expression/Gender and Wavy Hair/Gender tasks, the Pareto frontier of FairNVT is consistently shifted toward the desirable lower-right region, demonstrating that the proposed representation learning strategy better preserves task performance while improving fairness. On Big Nose/Age and Gender/Age tasks, FairNVT remains competitive, achieving comparable trade-offs to existing fairness-aware methods. This suggests that the proposed disentangled representation and targeted perturbation provide a more favorable balance between utility and fairness than applying fairness regularization directly to a single representation.

**Comparison at the best task accuracy.** While the Pareto frontiers characterize the complete trade-off of each method, practical deployment typically requires selecting a single model. Table 1 reports the model achieving the highest validation task accuracy for each method to complement this analysis with a representative operating point. Best results are shown in **bold**, and the second-best results are underlined. We report the mean and standard deviation of all evaluation metrics over three independent runs. For the baseline methods, the reported variability reflects randomness from model initialization and training. For FairNVT, it additionally includes the stochasticity introduced by sampling Gaussian noise during inference, thereby capturing both training and inference-time sources of randomness.

Across all four experiments, FairNVT consistently achieves comparable or higher task accuracy with competitive fairness metrics over the baselines. For example, on the Wavy Hair/Gender task, FairNVT attains the highest task accuracy of 87.8%, exceeding the best baseline by 2%, while simultaneously achieving the lowest EO, and EOpp violations. On UTKFace, FairNVT likewise maintains a high task accuracy of 98% together with consistently improved fairness metrics.

---

[6]FairVPT does not have official code release and is implemented by the authors based on the descriptions in the paper.

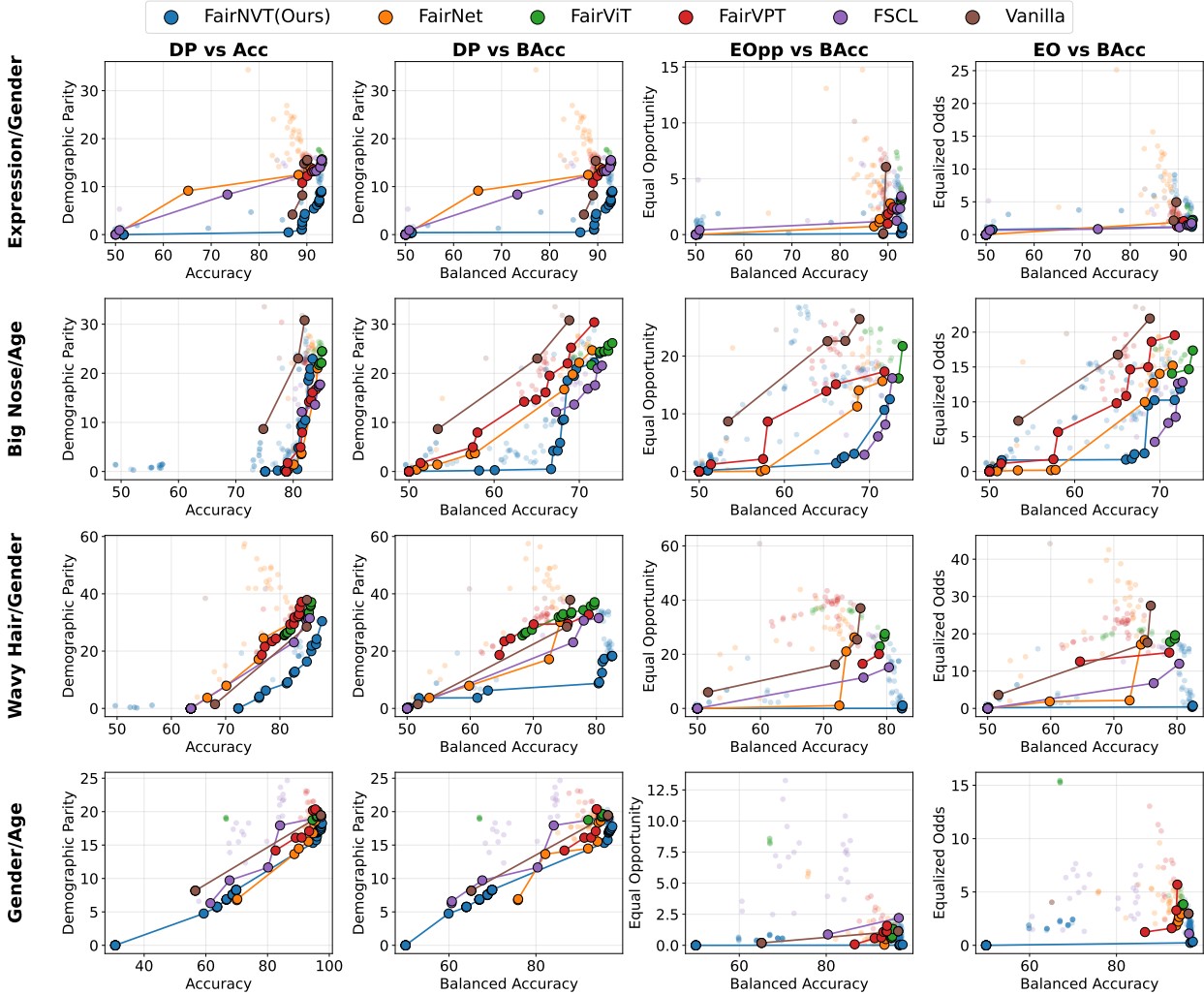

Figure 2: **Fairness–utility Pareto curves.** The lower-right region indicates higher task performance and lower fairness violations. FairNVT consistently achieves more favorable or competitive trade-offs than the baseline methods across the four task–sensitive attribute pairs.

**Qualitative results.** Figure 5 visualizes model attributions on *CelebA*, where task is *expression (smiling)* and sensitive attribute is *gender (male)*. We observe that the Vanilla model frequently relies on irrelevant background or on gender-correlated regions (e.g., hair/beard), while other baseline models down-weight such cues. Our method consistently attends more to expression-relevant regions such as the mouth, cheeks, and eyes, while effectively suppressing gender-related cues. This behavior aligns with the observed quantitative improvements in DP and EOpp, as well as the significant reduction in attacker accuracy.

## 4.2 Ablation Studies

**Effect of different model components.** We ablate the main components of FairNVT to evaluate their individual contributions to fairness and task performance. Table 2 summarizes the components included in each ablation together with the run selected based on the highest task accuracy. Figure 4 presents the Pareto curves across all hyperparameter configurations, illustrating the resulting fairness–utility trade-offs.

Comparing the Vanilla setup with Ablation 1 shows that introducing the fairness loss substantially reduces fairness violations on both datasets. For example, the DP difference decreases by 5.3 on Expression/Gender and by 8.6 on Big Nose/Age, indicating that directly optimizing the fairness objective is effective at improving

Table 1: **Image-Based Classification task:** Comparing our method with baselines on CelebA (a-c) and UTKFace (d) dataset. FairNVT demonstrates strong performance in higher task performance while achieving fair predictions.

| Method | Acc(↑) | BAcc(↑) | DP(↓) | EOpp(↓) | EO(↓) |
|---|---|---|---|---|---|
| Vanilla | 89.6(0.5) | 89.0(0.6) | 16.4(1.1) | 6.9(1.2) | 5.2(1.2) |
| ViT-FSCL | 92.9(1.0) | 92.6(1.0) | 15.5(2.2) | 3.5(1.2) | **1.8(1.1)** |
| FairViT | 93.1(0.2) | **92.8(0.3)** | 15.7(0.3) | 3.8(0.4) | 1.9(0.6) |
| FairVPT | 91.4(0.2) | 91.1(0.2) | 13.6(0.3) | 2.5(0.3) | 2.1(0.6) |
| FairNet | 90.7(0.3) | 90.5(0.4) | 13.9(0.4) | 2.8(0.2) | 1.9(0.4) |
| FairNVT(Ours) | **93.1(0.5)** | 92.8(0.5) | **9.1(1.8)** | **0.7(1.3)** | 2.5(1.5) |

(a) Task: Expression (Smiling), Sensitive Attribute: Gender (Male)

| Method | Acc(↑) | BAcc(↑) | DP(↓) | EOpp(↓) | EO(↓) |
|---|---|---|---|---|---|
| Vanilla | 81.2(0.8) | 67.9(1.2) | 30.8(1.2) | 26.4(0.2) | 22.0(0.2) |
| ViT-FSCL | **84.7(0.8)** | 69.1(1.6) | 17.7(2.4) | 17.2(2.1) | 11.5(2.8) |
| FairViT | 84.6(0.2) | **71.9(0.2)** | 24.5(0.8) | 22.5(1.4) | 16.7(1.2) |
| FairVPT | 83.1(0.2) | 66.0(0.3) | **16.1(1.0)** | **15.1(1.0)** | **11.0(0.8)** |
| FairNet | 84.2(0.3) | 68.0(0.2) | 21.0(0.5) | 20.4(0.8) | 15.2(0.7) |
| FairNVT(Ours) | 83.4(0.2) | 71.5(0.5) | 22.9(1.5) | 17.3(0.8) | 11.1(0.2) |

(b) Task: Big Nose, Sensitive Attribute: Age (Young)

| Method | Acc(↑) | BAcc(↑) | DP(↓) | EOpp(↓) | EO(↓) |
|---|---|---|---|---|---|
| Vanilla | 84.4(0.5) | 75.9(0.1) | 37.8(0.7) | 37.0(0.5) | 27.6(0.7) |
| ViT-FSCL | 85.5(0.5) | 77.8(2.7) | 31.5(1.1) | 26.4(2.1) | **16.1(2.6)** |
| FairViT | 85.8(0.4) | 79.7(0.3) | 37.1(0.8) | 27.5(1.1) | 19.6(0.9) |
| FairVPT | 84.0(0.4) | 74.8(0.4) | 37.2(0.8) | 38.1(0.8) | 24.9(1.0) |
| FairNet | 82.5(0.3) | 74.3(0.2) | **30.2(0.7)** | 27.1(0.6) | 17.1(0.9) |
| FairNVT (Ours) | **87.8(0.3)** | **80.6(0.2)** | 30.4(0.9) | **23.6(0.6)** | 17.1(0.7) |

(c) Task: Wavy Hair, Sensitive Attribute: Gender (Male)

| Method | Acc(↑) | BAcc(↑) | DP(↓) | EOpp(↓) | EO(↓) |
|---|---|---|---|---|---|
| Vanilla | 97.3(0.1) | 96.0(0.5) | 19.5(0.3) | 1.3(0.2) | 3.1(0.2) |
| ViT-FSCL | 97.4(0.2) | 96.7(0.5) | 19.2(0.1) | 2.2(0.3) | 1.1(0.1) |
| FairViT | 96.2(0.0) | 95.0(0.1) | 20.0(0.9) | 1.6(0.3) | 4.5(0.4) |
| FairVPT | 95.4(0.1) | 93.9(0.2) | 20.3(0.2) | 1.1(0.4) | 5.9(0.5) |
| FairNet | 95.5(0.3) | 94.9(0.3) | 18.6(0.2) | 0.6(0.8) | 2.9(0.4) |
| FairNVT(Ours) | **97.7(0.5)** | **97.4(0.5)** | 18.2(0.7) | **0.6(0.2)** | **0.9(0.7)** |

(d) Task: Gender, Sensitive Attribute: Age

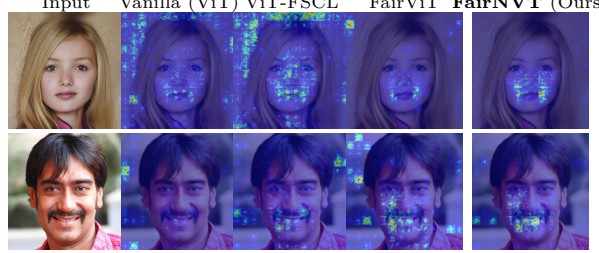 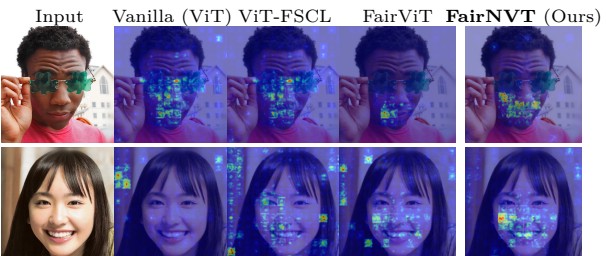

Figure 3: **Gradient-based saliency map** for the *Expression (smiling)* as main task and *Gender (male)* as sensitive attribute. Warmer regions indicate stronger contribution to the output logit. FairNVT primarily attends to expression-relevant areas (mouth/cheeks), demonstrating reduced reliance on gender-correlated cues.

downstream fairness. However, using only a single task adapter results in a larger utility cost, with task accuracy decreasing by 1.7% on Expression/Gender and 2.7% on Big Nose/Age compared with the full FairNVT model.

Comparing Ablations 1 and 2, Ablations 1, 2 and 3 highlights the contribution of the split-adapter architecture and the orthogonality loss respectively. Introducing a separate Sensitive Adapter (Ablation 2) improves task accuracy by 1.1% on Expression/Gender and 2.3% on Big Nose/Age relative to Ablation 1, while adding the orthogonality loss (Ablation 3) provides a further gain of 0.3% and 0.2%, respectively. These improvements

Table 2: **Effect of model components.** Each ablation adds a component of the proposed method. Results correspond to the hyperparameter configuration achieving the highest task accuracy for each variant.

| Ablation | Adapter | Fair Loss | Orth Loss | Noise | Expression/Gender Acc(↑) | BAcc(↑) | DP(↓) | EOpp(↓) | EO(↓) | Big Nose/Age Acc(↑) | BAcc(↑) | DP(↓) | EOpp(↓) | EO(↓) |
|---|---|---|---|---|---|---|---|---|---|---|---|---|---|---|
| Vanilla | Task only | ✗ | ✗ | ✗ | 89.6 | 89.0 | 16.4 | 6.9 | 5.2 | 81.2 | 67.9 | 30.8 | 26.4 | 22.0 |
| Abl-1 | Task only | ✓ | ✗ | ✗ | 91.4 | 91.2 | 11.1 | 1.1 | 1.4 | 80.7 | 70.1 | 22.2 | 11.6 | 9.9 |
| Abl-2 | Task + Sensitive | ✓ | ✗ | ✗ | 92.2 | 91.6 | 14.4 | 5.8 | 3.3 | 83.0 | 75.4 | 27.3 | 12.2 | 13.0 |
| Abl-3 | Task + Sensitive | ✓ | ✓ | ✗ | 92.5 | 91.8 | 12.3 | 4.4 | 2.5 | 83.2 | 74.7 | 27.2 | 13.5 | 13.5 |
| FairNVT | Task + Sensitive | ✓ | ✓ | ✓ | 93.1 | 92.8 | 9.1 | 0.7 | 2.5 | 83.4 | 71.5 | 22.9 | 17.3 | 11.1 |

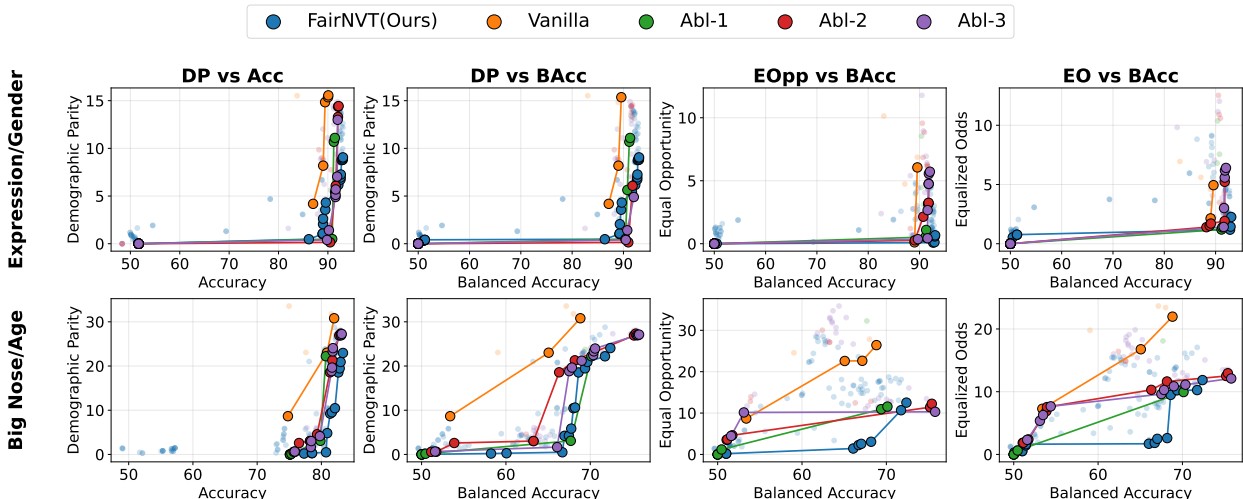

Figure 4: **Fairness–utility trade-off of the ablation variants.** Pareto curves over all hyperparameter configurations. The lower-right region indicates better trade-offs, corresponding to higher task performance and lower fairness violations. FairNVT consistently lies on or near the Pareto frontier, demonstrating the most favorable overall trade-off.

are accompanied by higher fairness violations than Ablation 1, suggesting that the additional sensitive branch introduces strong sensitive information signal that is also useful for the downstream task. The orthogonality loss partially mitigates this effect, reducing DP by 2.1 and 0.1 on Expression/Gender and Big Nose/Age, respectively, comparing with Ablation 2.

Finally, the full FairNVT model has the most favorable overall fairness–utility trade-off by combining the split-adapter architecture, orthogonality regularization, noise injection, and fairness optimization. On Expression/Gender, FairNVT achieves both the highest task accuracy and the lowest fairness scores among all ablations. On Big Nose/Age, Ablations 2 and 3 can achieve higher task accuracy, as shown by the rightmost points in Figure 4 (Row 2), but these gains come with substantially larger fairness violations. In contrast, FairNVT maintains competitive task performance while achieving improved DP scores.

**Evaluate sensitive attribute leakage with attacker.** We evaluate the amount of sensitive information retained in different learned representations by training an attacker to predict the sensitive attribute from each embedding. Specifically, after training FairNVT, we extract the task embedding ($e_t$), sensitive embedding ($e_s$), noised sensitive embedding ($e_s^{\text{noised}}$), and the fused embedding ($e_f$), where the embedding weights are frozen, and a separate attacker is trained on each embedding to predict the sensitive attribute.

As shown in Table 3 (Column 1-4), the sensitive embedding ($e_s$) has high or nearly perfect prediction accuracy ($\approx 99\%$ in Expression/Gender), indicating that the sensitive adapter successfully captures information related to the sensitive attribute. In contrast, the task embedding ($e_t$) yields substantially lower attacker performance ($\approx 80\%$ in Expression/Gender), suggesting that the split-adapter architecture together with the orthogonality loss reduces, though does not eliminate, sensitive information in the task embedding. Applying Gaussian noise to the sensitive embedding ($e_s^{\text{noised}}$) reduces attacker performance to nearly random guessing ($\approx 50\%$ in Expression/Gender), demonstrating that noise substantially suppresses sensitive information in this channel. The attacker performs better on the fused embedding ($e_f$) than on $e_s^{\text{noised}}$ alone, but worse than on the task embedding $e_t$. These results indicate that noise suppresses the sensitive information in the sensitive channel, and the residual sensitive information can still exist in the task information channel, where the fairness loss handles it.

Since the task embedding still contains residual sensitive information, we further evaluate stronger attacker models with additional hidden layers. As shown in Table 3 (Column 5-6), attacker accuracy increases only gradually by around 1% as model capacity grows from 3 to 10 hidden layers. This suggests the sensitive

information is more recoverable but still substantially less predictable, which is consistent with the intended role of the split-adapter architecture and targeted perturbation.

We additionally evaluate the case where the fused representation concatenates the task embedding with the average of multiple independently perturbed sensitive embeddings in Table 3 (Column 7-8). The attacker performance increases as expected, as averaging over $k$ draws of independent Gaussian noise reduces the variance to $\sigma^2/k$, where $\sigma^2$ is the variance of a single noise draw. The reduced variance makes the sensitive information easier to recover.

**Summary of ablation study.** Taken together, *the component ablation and attacker analysis provide empirical support for the design of FairNVT.* The split-adapter architecture creates a dedicated pathway for sensitive information, enabling targeted perturbation of the sensitive representation, while the orthogonality loss encourages task and sensitive adapters to be learn more decoupled representations. Gaussian perturbation substantially suppresses sensitive information in the sensitive pathway before fusion, whereas the fairness loss discourages the downstream classifier from relying on the remaining sensitive information present in the fused representation. Collectively, these components produce representations with reduced sensitive attribute exposure while preserving task-relevant information, leading to improved fairness–utility trade-offs across the evaluated experiments.

Table 3: **Sensitive attribute leakage measured by attacker performance.** Attacker accuracy and balanced accuracy on different learned representations. Additional columns evaluate stronger attacker models (Att3, Att10 represent a MLP attacker with 3, 10 hidden layers respectively), and repeated observations of the noised sensitive embedding through averaging multiple perturbations (Avg5, Avg50 represent averaging over 5, 50 randomly perturbed sensitive embedding respectively).

|  |  | $e_t$ | $e_s$ | $e_s^{\text{noised}}$ | $e_f$ | $e_f$ w/Att3 | $e_f$ w/Att10 | $e_f$ w/Avg5 | $e_f$ w/Avg50 |
|---|---|---|---|---|---|---|---|---|---|
| **Expression/Gender** | **Att.Acc** | 68.3 | 99.0 | 51.7 | 52.1 | 52.2 | 52.4 | 63.6 | 92.5 |
|  | **Att.BAcc** | 68.6 | 99.0 | 50.2 | 51.7 | 50.8 | 51.1 | 62.0 | 92.3 |
| **Big Nose/Age** | **Att.Acc** | 74.6 | 88.2 | 63.1 | 67.6 | 67.9 | 68.3 | 77.1 | 85.9 |
|  | **Att.BAcc** | 60.4 | 80.1 | 50.1 | 53.2 | 53.3 | 53.6 | 59.7 | 76.3 |

## 5 Conclusion

We introduced FairNVT, a plug-in framework for fair classification with frozen backbone models. By learning separate task and sensitive representations, injecting calibrated Gaussian noise into the sensitive pathway, and jointly optimizing orthogonality and fairness objectives, FairNVT achieves favorable fairness–utility trade-offs across multiple experiments, reducing fairness violations while maintaining competitive task performance.

FairNVT has several limitations. First, similar to many supervised fairness methods, our method requires sensitive attribute labels during training. This assumption may limit applicability in settings where sensitive labels are unavailable, unreliable, undesirable to collect or severely imbalanced. Second, although inference is reproducible with fixed random seeds, FairNVT relies on stochastic perturbations and therefore produces stochastic predictions. The task embedding is deterministic but may not be reliably reusable as a debiased embedding for other downstream tasks. Finally, our design is motivated by the intuition of separating task and sensitive information, but does not guarantee complete disentanglement of the learned representations.

These limitations motivate several directions for future work. Developing stronger objectives for separating task and sensitive information and providing a deeper theoretical understanding of the proposed framework are promising directions. Designing stronger fairness loss for severe class imbalance cases where achieving fairness is more difficult. In addition, while our experiments focus on image and text classification with transformer encoders, the framework is largely architecture agnostic and can naturally be extended to other modalities and transformer-based models. We hope the proposed framework encourages future work on broader fairness tasks beyond binary classification, including multi-class, multi-attribute, and broader scenarios such as vision-language models.

## Broader Impact Concerns

This work uses face datasets with perceived gender and age labels, including binary groupings. Although this is standard in the benchmark datasets and existing literature, these labels can be ethically sensitive, noisy and potentially harmful if used outside controlled research settings. We also clarify that lower benchmark disparity as reported in this work does not certify fairness in real deployments, especially under distribution shift or for unmodeled intersectional groups.

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

# A   Related Works

**Transformers for Classification.**   Transformers have seen broad adoption for classification in both vision and text. Image-level transformer models such as ViT Dosovitskiy et al. (2021) have been widely used across domains including face analysis Dan et al. (2023); Narayan et al. (2025); Jacob & Stenger (2021), medical imaging Shao et al. (2021); Tang et al. (2022), and general object recognition Khan et al. (2022); Wang et al. (2025). These backbones match or surpass strong CNN models while offering flexible transfer to new datasets.

Similarly, transformer-based language models (e.g., BERT Devlin et al. (2019), RoBERTa Liu et al. (2019), DeBERTa He et al. (2021)) have become the dominant choice for text classification, often outperforming CNN/RNN architectures and transferring effectively via pretrain–adapt pipelines.

Given their importance, understanding and mitigating their fairness challenges is crucial. We focus on image classification with frozen vision transformers and show that the proposed framework also transfers effectively to text.

**Fairness Approaches.**   Many approaches mitigate group disparities by modifying the training distribution itself. Classical methods include reweighing Kamiran & Calders (2012), disparate impact removal Feldman et al. (2015), and optimized preprocessing Calmon et al. (2017), which explicitly adjust sample weights or features to balance sensitive groups. More recent strategies alter the training data more subtly through curated fine-tuning Ghanbarzadeh et al. (2023), group rebalancing, or fairness-oriented augmentation Sun et al. (2023); Halevy et al. (2025), aiming to reduce distributional bias without modifying model parameters.

Unlike data modification approaches, we make no data level changes, group labels are used only at training to optimize demographic parity, and not required at inference.

Beyond data manipulation, fairness has also been pursued through changing the learning objective Agarwal et al. (2018); Zhang et al. (2018). More recently, transformer-based methods adjust attention Zhou et al. (2024), mask bias-correlated ViT regions Tian et al. (2024), or apply fairness-aware prompting Park & Byun (2024). These recent approaches typically adjust attention or prompting, whereas we operate in a learned sensitive latent subspace and apply randomized smoothing without architectural changes or retraining the frozen model.

Another direction introduces parameter-efficient modules for debiasing, such as adapters Fatemi et al. (2023); Hauzenberger et al. (2023); Yang et al. (2023); Lauscher et al. (2021); Kumar et al. (2023); Masoudian et al. (2024). For example, DAM Kumar et al. (2023) adds debiasing adapters alongside task adapters to handle multiple sensitive attributes, while ConGater Masoudian et al. (2024) introduces controllable gates that balance fairness and utility at inference time. Although these approaches lower training cost, they typically act indirectly on representations without explicitly identifying or perturbing a sensitive subspace. In contrast, our method keeps the transformer backbone frozen and directly manipulates a learned sensitive subspace through noise injection.

Recent methods improve fairness by directly altering latent representations, with approaches based on latent factorization or variational modeling Zemel et al. (2013); Louizos et al. (2015) and adversarially aligned representations Madras et al. (2018); Zhang et al. (2018); Götte (2023) that aim to reduce sensitive information in learned features through min–max optimization, can be unstable and often requiring multi-stage training.

Projection-based methods such as INLP Ravfogel et al. (2020), sufficient projection (SUP) Shi et al. (2024), and SLSD Islam et al. (2024) remove subspaces predictive of sensitive attributes; however, linear removal can discard task-relevant information when sensitive and semantic directions overlap. Information-theoretic

approaches Kang et al. (2022); Wang et al. (2023); Xie et al. (2024) estimate mutual information to regularize fairness, while contrastive debiasing Park et al. (2022); Shen et al. (2021) often relies on group-balanced sampling and two-stage optimization. Recent concept-editing methods Karvonen et al. (2024) learn sparse subspaces aligned with sensitive concepts and suppress them to reduce probe recoverability.

**Fairness via Smoothing Models.** Randomized smoothing Lecuyer et al. (2019); Cohen et al. (2019) is primarily studied as a robustness technique, where prediction stability under noise yields certified guarantees for robust predictions. Although not originally developed for fairness, the resulting invariance suggests that smoothing could help reducing reliance on sensitive factors.

Individual fairness is formalized via task-relevant similarity metrics Dwork et al. (2012). Empirical work connecting smoothing to fairness remains limited. For example, Jin et al. (2022) trains group-specific models and averages their parameters to certify group fairness in low-dimensional tabular settings, while Yeom & Fredrikson (2021); Peychev et al. (2022) encourage individual fairness via smoothing in input or latent spaces. These approaches, however, require isolated sensitive attributes in tabular data style, or operate only when input perturbations are well defined. Unlike prior works, our method performs smoothing selectively in a learned sensitive subspace, suppressing sensitive variation while preserving task structure, without architectural changes or using sensitive labels at inference.

# B  Obfuscating Sensitive Information Improves Fairness

In this section, we demonstrate the mathematical intuition to the design of the FairNVT framework.

Let $(X, Y)$ denote the feature-label pair, where the features $X$ may contain information about both the task label $Y$ and a sensitive attribute $S$. Let $Z = e(X)$ denote the representation produced by an encoder $e$, for example, the pre-trained frozen backbone models, and let $h$ be a classifier producing predictions $\hat{Y} = \mathbb{1}(h(Z) > \tau)$ with a threshold $\tau$. In the case where both $S, Y$ are binary attributes, we define demographic parity difference as $DP := |\Pr(\hat{Y} = 1|S = 0) - \Pr(\hat{Y} = 1|S = 1)|$, then the following result is an immediate consequence of the definition of total variation distance.

**Proposition B.1.** *If $Z \perp\!\!\!\perp S$, then $DP = 0$.*

*Proof.* Let $A$ be the event that the classifier $h$ predicts $\hat{Y} = 1$, i.e. $A = \{z : \mathbb{1}(h(z) > \tau) = 1\}$, and let $P, Q$ be the conditional distribution of $Z|S = 0$ and $Z|S = 1$. At inference time $\hat{Y}$ is a deterministic function of $Z$, so $P(A) = \Pr(Z \in A|S = 0) = \Pr(\hat{Y} = 1|S = 0)$, $Q(A) = \Pr(\hat{Y} = 1|S = 1)$. By the definition of demographic parity difference ($DP$) and total variation distance ($\delta_{TV}$),

$$DP := |P(\hat{Y} = 1|S = 0) - P(\hat{Y} = 1|S = 1)| = |P(A) - Q(A)| \leq \sup_A |P(A) - Q(A)| := \delta_{TV}(P, Q).$$

If $Z \perp\!\!\!\perp S$, then $P = Q$ and $\delta_{TV}(P, Q) = 0$ hence $DP = 0$. $\qquad\square$

Although exact independence between $Z$ and $S$ is difficult to achieve in practice, reducing the dependence of the learned representation on the sensitive attribute is a natural surrogate objective. The following proposition describes an *idealized* setting in which the sensitive component of the representation is completely randomized and the task embedding is made independent of the sensitive attribute.

**Proposition B.2.** *Suppose the learned representation allows a decomposition $Z = (Z^t, Z^s)$, where $Z^t$ contains task-relevant non-sensitive information and satisfies $Z^t \perp\!\!\!\perp S$. Let $N$ be a random variable independent of both $S$ and $Z^t$. If the sensitive component $Z^s$ is replaced by $N$, then the resulting representation $\tilde{Z} = (Z^t, N)$ satisfies $\tilde{Z} \perp\!\!\!\perp S$.*

*Proof.* Since $Z^t \perp\!\!\!\perp S$ and $N$ is independent of both $Z^t$ and $S$, $P(\tilde{Z}, S) = P(Z^t, N, S) = P(Z^t)P(N)P(S) = P(\tilde{Z})P(S)$, which implies $\tilde{Z} \perp\!\!\!\perp S$. $\qquad\square$

The propositions illustrate the intuition behind our framework. While FairNVT does not replace the sensitive component with an independent random variable, it injects Gaussian noise into the sensitive embedding to

progressively reduce the information it carries about the sensitive attribute while preserving task-relevant information. On the other hand, we apply orthogonality constraint to decoupling task and sensitive embedding, encouraging $Z^t$ to be less correlated with $S$. Even though achieving independence is challenging, these operations move toward the idealized setting to reduce sensitive information exposure to the downstream task classifier. Consequently in the ideal case, we expect the representations of different sensitive groups to become less distinguishable. Since the demographic parity difference is upper bounded by the total variation distance between $P(Z \mid S = 0)$ and $P(Z \mid S = 1)$, we expected reduced demographic parity differences. Moreover, since total variation distance upper bounds the optimal accuracy of distinguishing between two probability distributions (Theorem 3.1 of Aubinais et al. (2023)), reducing the discrepancy between sensitive group embeddings also limits the accuracy of predicting the sensitive attribute from the learned representation.

For Equalized Odds (EO) and Equal Opportunity (EOpp), a stronger condition $Z \perp\!\!\!\perp S|Y$ is sufficient to ensure fairness when predictions depend only on the learned representation. While our framework does not explicitly enforce this conditional independence, reducing sensitive information in the learned representation may also reduce conditional disparities in practice, which we verify empirically.

## C  Experiment Setup Details

**Model Architectures**   We use the ViT-Base model [7] as the frozen backbone for the CelebA and UTKFace datasets. Both task and sensitive adapters are bottleneck adapters inserted into each Transformer block of the frozen backbone. Each adapter consists of a down-projection that maps hidden states to a lower-dimensional space and an up-projection that restores them to the original hidden dimension. The reduction factor is a tunable hyperparameter. The task and sensitive classification heads are Multi-Layer Perceptrons (MLPs) whose hidden layer size matches the respective embedding dimension; the number of hidden layers is also treated as a tunable hyperparameter. For evaluating representation-level fairness via attacker accuracies, we use an attacker network with the same architecture as the task classification head. Table 4 summarizes an example FairNVT architecture used in the experiment for the task attribute *expression (smiling)* and the sensitive attribute *gender (male)*. FairNVT for vision task trains only 5.4M parameters ($\sim 6\%$ of ViT-Base) by freezing the backbone and introducing lightweight adapters and classification heads, significantly reducing computational cost compared to full fine-tuning.

Table 4: FairNVT architectural specifications for the experiment with task attribute *expression (smiling)* and sensitive attribute *gender (male)*. Layer dimensions are denoted as $N_{\text{weight\_in}} \times N_{\text{weight\_out}} + N_{\text{bias}}$. Task and sensitive adapter layers are attached after the final dense layer of the frozen ViT encoder (output dimension = 768) in all 11 encoder layers. Noise injection and embedding concatenation introduce no trainable parameters.

| Architecture | Layer | Specification | Output Size |
|---|---|---|---|
| Task Adapter | down_projection | $(768 \times 96 + 96) \times 11$ | 96 |
| | up_projection | $(96 \times 768 + 768) \times 11$ | 768 |
| Sensitive Adapter | down_projection | $(768 \times 48 + 48) \times 11$ | 48 |
| | up_projection | $(48 \times 768 + 768) \times 11$ | 768 |
| Noise Injection | \ | \ | 768 |
| Embedding Concatenation | \ | \ | $768 \times 2$ |
| Task Clf Head | linear_0 | $(768 \times 2) \times (768 \times 2) + (768 \times 2)$ | $768 \times 2$ |
| | tanh_activation | \ | $768 \times 2$ |
| | linear_1 | $(768 \times 2) \times 2 + 2$ | 2 |
| Sensitive Clf Head | linear_0 | $768 \times 768 + 768$ | 768 |
| | tanh_activation | \ | 768 |
| | linear_1 | $768 \times 2 + 2$ | 2 |

**Implementation Details.**   We implement all models in PyTorch Paszke et al. (2019) and train them on a workstation equipped with an AMD EPYC 7H12 CPU (64 cores) with a NVIDIA A100 GPU. For both our

---

[7]`google/vit-base-patch16-224`

method and the baselines, we train the models using AdamW (Loshchilov & Hutter, 2019; Kingma & Ba, 2015) with batch size 256 and default hyper parameters of $\beta_1 = 0.9$, $\beta_2 = 0.999$, a weight decay of 0.01, and a batch size of 256. The adapter architecture uses a reduction factor of 8 for the task branch and 16 for the sensitive branch. Training the debiasing framework for one run takes approximately 3 hours on a single A100 GPU. Inference requires 1.1 seconds for a batch of 256 samples.

During training, we run a grid search over other sensitive hyperparameters including learning rates and loss weights, and report the best validation-selected results. Specifically, we perform grid-search hyperparameter tuning over the following ranges: adapter reduction factor $\{4, 8, 16\}$; number of hidden layers $\{0, 1, 2\}$; learning rates (searched by orders of magnitude, e.g., $1e-1$, $1e-2$, etc., until the best run is not at a boundary value); gradient-clipping thresholds $\{1, 10\}$; noise levels $\{1, 5, 10\}$; and loss-weight coefficients $\beta \in \{0.1, 0.3, 0.5, 1.0\}$.

We tune the following hyperparameters for the baseline methods using validation performance for model selection. For FairVPT, we grid search the fairness loss weight $\lambda \in \{0.01, 0.1, 0.5, 1.0\}$ and the phase-two linear probing learning rates, while fixing the number of cleaner prompts to 25 as recommended by the original implementation. For FairViT, we tune the learning rates, the distance loss weight $\alpha \in \{0.1, 0.5, 1.0, 5.0\}$, and the mask optimization step size $\{0.1, 1.0, 3.0\}$. For FSCL, we tune the phase-one and phase-two learning rates, the contrastive temperature $\{0.05, 0.1, 0.3\}$. For FairNet, we tune the learning rate, and the weight parameters $\lambda_c$ and $\lambda_d$ in $\{0.1, 0.5, 1.0, 5.0\}$.

For evaluation, accuracy and balanced accuracy are computed from the predicted and true task labels. Fairness metrics (DP, EO, and EOpp) are computed using the predicted and true task labels together with the true sensitive attributes. The attacker setup follows Kumar et al. (2023): in an independent run, the attacker receives the task-classifier embeddings as input $X$ and the corresponding sensitive attributes $Y$ from the training and test sets. The attacker is trained to predict $Y$ from $X$ until the training accuracy no longer improves significantly, and its test accuracy is reported as the attacker's ability to recover the sensitive attribute from the learned representation.

## D   More Experiment Results

### D.1   Image-based Classification Results

**More experiments on CelebA.**   Table 13 shows the results on more task, sensitive attribute pairs in the CelebA dataset. In most cases, we observe FairNVT showing a good balance between the prediction and fairness objectives, achieves better or comparable performances to the best baseline across different metrics.

**Additional qualitative results on CelebA.**   We provide additional samples for the *expression (smiling)* task with *gender (male)* as the sensitive attribute. Heatmaps are computed with *SmoothGrad* on the predicted class logit by averaging input gradients over 25 Gaussian noised, normalized inputs ($\sigma = 0.10$), aggregating $|\nabla_x|$ across channels, bilinearly resizing, and applying a light $3 \times 3$ blur, with maps normalized independently per image so intensities reflect within panel variation. Warmer regions indicate stronger contributions to the output logit. FairNVT concentrates on expression-relevant areas (e.g., mouth, cheeks), suggesting reduced reliance on gender-correlated cues and improved fairness via task-specific evidence.

### D.2   Text-based Classification Experiments and Results

**Text-based task.** We additionally evaluate on **text-based classification tasks** with Bert-Base as the frozen backbone to compare with baselines Kumar et al. (2023); Ravfogel et al. (2020). The dataset BIOS De-Arteaga et al. (2019) consists of professional biographies with occupation labels, and the task is to predict occupation while evaluating fairness with respect to perceived gender.

**Multi-class fairness loss.** Since the task (*Profession*) in BIOS is a multi-class attribute, we extend the fairness loss in Eq. 3 to aggregate disparity with respect to the sensitive attribute across all task classes: let $n_0, n_1$ be the number of samples in a batch belonging to sensitive group $0, 1$ respectively, $p_k = p_\theta(\hat{y} = k|x)$ be

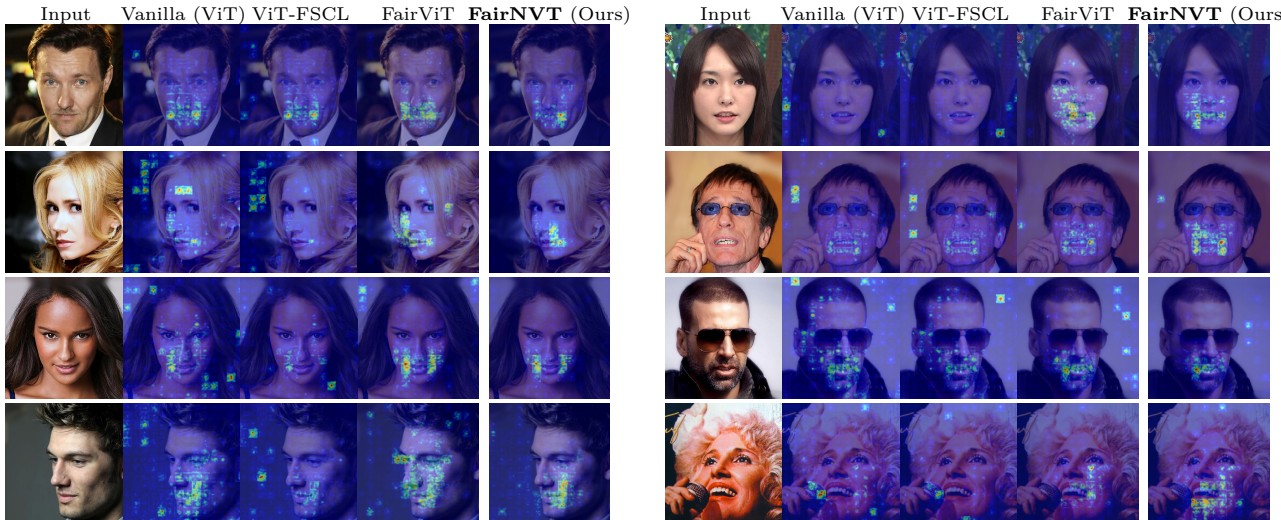

Figure 5: Additional examples: Gradient-based saliency map for the *expression (smiling)* as main task and *gender (male)* as sensitive attribute. Warmer regions indicate stronger contribution to the output logit. FairNVT primarily attends to expression-relevant areas (mouth/cheeks), demonstrating reduced reliance on gender-correlated cues.

the probability of predicting class $k$ of label $y$, and $\mathbf{1}[\cdot]$ be the indicator function then,

$$L_{\text{dp}}^{\text{multi}}(\theta) = \sum_{k=1}^{c} \left| \frac{1}{n_0} \sum_{i=1}^{n_0} \mathbf{1}[s_i = 0]p_{i,k} - \frac{1}{n_1} \sum_{j=1}^{n_1} \mathbf{1}[s_j = 1]p_{j,k} \right|.$$

The overall loss remains the same form except substituting in the multi-class fairness loss, $L = L_{\text{ce}}^{\text{t}} + \beta_1 L_{\text{ce}}^{\text{s}} + \beta_2 L_{\text{orth}} + \beta_3 L_{\text{dp}}^{\text{multi}}$, where $\beta$s are hyperparameters representing weights on each loss.

**Text-based fair classification baselines.** We include the Vanilla setup and five baselines:

- **Vanilla-BERT** Devlin et al. (2019): Standard fine-tuning without fairness intervention.

- **FT-Debias** Kumar et al. (2023): Fine-tuning with adversarial debiasing objectives.

- **INLP** Ravfogel et al. (2020): Iteratively trains linear probes on the sensitive attribute and projects embeddings to remove the corresponding subspaces.

- **SUP** Shi et al. (2024): Projection-based concept removal that preserves task-relevant features while suppressing sensitive directions.

- **ConGater** Masoudian et al. (2024): Group-aware contrastive training to disentangle task and sensitive representations.

- **DAM** Kumar et al. (2023): Parameter-efficient debiasing using adapter fusion to reduce demographic leakage.

**Comparison on BIOS.** Table 5 reports results on BIOS De-Arteaga et al. (2019), where the task is multi-class *Profession*[8] and the sensitive attribute is *Gender*. We report the mean results and its standard deviation over 3 runs. Overall, FairNVT offers a favorable fairness–utility trade-off, pairing competitive

---

[8]EO and EOpp are defined to condition on the true task label and takes the difference between values conditioned on opposite binary task labels. We report DP in multi-class settings since DP does not condition on the true task label and remains applicable with its original form of definition.

accuracy with state-of-the-art DP and sensitive-attribute leakage close to the best baseline. Figure 6 compares logits for original and gender-swapped sentences, where pronouns are replaced with those of the opposite gender. FairNVT produces more similar distributions between the two, indicating reduced sensitivity to gender. Additional details, illustrative examples, and predicted scores are provided in the supplementary materials. These results highlight that FairNVT can effectively extend from the vision domain to textual embeddings.

Table 5: **Text-Based Classification task:** Comparing our method with baselines on Bios De-Arteaga et al. (2019) dataset, Task: Profession (Multi-Class), Sensitive Attribute: Gender. All reported values are scaled by $\times 10^2$.

| Method | Acc($\uparrow$) | DP($\downarrow$) |
|---|---|---|
| **Vanilla-BERT** | $72.8_{(0.2)}$ | $2.0_{(0.2)}$ |
| **+FT-Debias** | $76.8_{(2.4)}$ | $2.1_{(0.2)}$ |
| **+INLP** | $76.4_{(0.1)}$ | $\underline{1.7_{(0.1)}}$ |
| **+SUP** | $77.2_{(0.2)}$ | $2.1_{(0.5)}$ |
| **+DAM** | $80.3_{(0.4)}$ | $2.2_{(0.5)}$ |
| **+CONGATER** | $\mathbf{82.4_{(0.5)}}$ | $1.9_{(0.3)}$ |
| **+FairNVT(Ours)** | $\underline{80.6_{(0.4)}}$ | $\mathbf{1.6_{(0.1)}}$ |

Figure 6: **Robustness to gender-indicator swapping on BIOS.** We plot the distribution of the model's confidence in predicting profession for the original text and its gender-swapped counterpart for 100 random samples. FairNVT (right) exhibits more overlapping distributions than Vanilla (left) in more confident predictions.

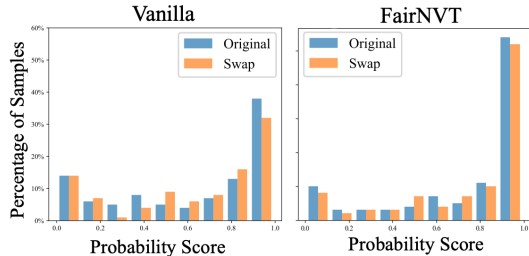

**Qualitative results on BIOS.** We evaluate fairness by comparing predictions on pairs of sentences that are identical except for words that indicate gender. Table 14 summarizes how the predicted profession probabilities change under these minimal substitutions. The vanilla model shows substantial shifts, whereas FairNVT produces more stable predictions across sentences that differ only in gender-indicative terms.

## E  More Ablation Results

If not specified specifically, the ablation studies is performed on task attribute: Expression (Smiling), sensitive attribute: Gender (Male) task using the CelebA dataset.

**Sensitivity to batch size.** The proposed fairness loss estimates group-level statistics within each mini-batch and may therefore become less reliable when the batch size is small, particularly when the sensitive attribute is imbalanced. Smaller batches also increase the variance of the optimization process more generally. This section evaluates the sensitivity of FairNVT to the training batch size. Table 6 compares the performance of FairNVT using batch sizes of 256, 128, and 64.

For the Gender attribute, where the two sensitive groups are relatively balanced (57% versus 43% of the training samples), the performance remains largely unchanged as the batch size decreases. For the Age attribute, where the group distribution is more imbalanced (77% versus 23%), reducing the batch size leads to a noticeable drop in task performance. Although the fairness metrics improve in this setting, the improvement is likely influenced by the degraded task accuracy, as predictions become closer to random guessing and therefore exhibit smaller group disparities.

We note that when hardware constraints limit the feasible batch size, we use techniques such as virtual batching to improve optimization stability by approximating updates with a larger effective batch size. Specifically, forward and backward passes are performed on multiple micro-batches and their gradients are accumulated. The model parameters are updated only after the accumulated gradients correspond to the desired effective batch size. While the fairness loss is still computed independently on each micro-batch and

therefore rely on estimates of the sensitive-group statistics from smaller batches, gradient accumulation help reduce the variance of the overall parameter updates.

Table 6: **Sensitivity of batch size.** We ablate on the batch size when training on datasets with mini-batches in case the sensitive attribute is imbalanced. The Gender and Age attribute has majority/minority groups having 57%/43%, 77%/23% samples respectively.

| | Batch Size | Acc(↑) | BAcc(↑) | DP(↓) | EOpp(↓) | EO(↓) | Att.Acc(↓) |
|---|---|---|---|---|---|---|---|
| **Task: Expression (Smiling)** | 256 | 93.1 | 92.8 | 9.1 | 0.7 | 2.5 | 52.1 |
| **Sens.: Gender (Male)** | 128 | 93.1 | 93.0 | 9.8 | 0.2 | 1.5 | 53.3 |
| | 64 | 92.6 | 92.7 | 8.5 | 1.2 | 2.7 | 52.8 |
| **Task: Big Nose** | 256 | 83.4 | 71.5 | 22.9 | 17.3 | 11.1 | 67.6 |
| **Sens.: Age (Young)** | 128 | 82.0 | 70.7 | 12.3 | 3.8 | 3.4 | 69.3 |
| | 64 | 81.5 | 71.3 | 14.8 | 4.0 | 4.1 | 68.8 |

**Variation in performance from inference-time randomness.** The standard deviations reported for the main results reflect variability from both training and inference, as the entire training and evaluation pipeline is repeated multiple times using the selected hyperparameter configuration. This section isolates the variability arising from inference-time stochasticity only. Specifically, we fix the trained model at the checkpoint obtained using the best hyperparameter configuration and repeat inference three times, where the only source of randomness is the Gaussian noise sampled during inference.

Table 7 reports the resulting inference-time variability. We observe that the standard deviations are comparable to those obtained when both training and inference are repeated, indicating that FairNVT produces stable predictions when deployed with a fixed trained model. These results suggest that the stochastic noise injection at inference introduces only modest variability.

Table 7: **Variation in performance from inference-time randomness.** We examine the variation in performance when the training model parameters are fixed and different random noise is drawn at inference time to make predictions.

| | Acc(↑) | BAcc(↑) | DP(↓) | EOpp(↓) | EO(↓) | Att.Acc(↓) |
|---|---|---|---|---|---|---|
| **Task: Expression (Smiling) Sens.: Gender (Male)** | $93.1_{(0.06)}$ | $92.9_{(0.07)}$ | $9.4_{(0.15)}$ | $1.1_{(0.23)}$ | $1.5_{(0.07)}$ | $53.6_{(0.03)}$ |
| **Task: Big Nose Sens.: Age (Young)** | $83.2_{(0.12)}$ | $70.8_{(0.31)}$ | $15.8_{(0.26)}$ | $8.3_{(0.43)}$ | $7.7_{(0.25)}$ | $68.5_{(0.03)}$ |

**Analysis of noise strength ($\sigma$).** We analyze how the noise strength $\sigma$ influences both representation and prediction level fairness. As shown in Table 8, moderate noise substantially lowers attacker accuracy, indicating that the injected perturbation effectively hides sensitive information without disturbing the task signal. When the noise becomes very large, the model shows improvements in several fairness metrics (DP, EO, Att.Acc) but shows a decline in accuracy. In practice, a moderate noise level provides a stable trade-off between privacy and utility.

Table 8: **Sensitivity to noise.** We ablate on noise levels for *expression (smiling)* as main task and *gender (male)* as sensitive attribute. Moderate noise levels balance utility (Acc/BAcc), fairness gaps (DP/EOpp/EO), and sensitive information leakage (Att. Acc). Very large noise further improves most fairness metrics but begins to slightly reduce accuracy, reflecting a utility-fairness trade-off at higher noise levels.

| Noise Level ($\sigma$) | Acc(↑) | BAcc(↑) | DP(↓) | EOpp(↓) | EO(↓) | Att.Acc(↓) |
|---|---|---|---|---|---|---|
| 1 | 93.0 | 93.1 | 9.4 | 1.0 | 2.0 | 67.4 |
| 5 | 93.1 | 92.8 | 9.1 | 0.7 | 2.5 | 51.7 |
| 100 | 91.0 | 91.2 | 9.2 | 0.9 | 1.1 | 50.5 |

**Comparing different task classifier inputs.** Table 9 shows the effect on accuracy and fairness metrics when the task classifier input changes. These results are trained with the same loss $L$ as in Section 3, except

nullifying the orthogonality loss when the sensitive embedding ($e^s$) is not present. When using the backbone frozen embedding $h$ directly (row 1) or concatenating task ($e_t$) and sensitive embedding ($e_s$) without noise (row 2), it is more difficult to obtain fair outcomes when the sensitive information is not obfuscated, indicated by higher DP, EOpp, EO and Att.Acc values. Naively adding noise to $h$ (row 3) could achieve better fairness outcomes but hurting the task performance with 3.8% decrease in accuracy. This indicate that separating into dedicated task and sensitive pathways is preliminary and favorable for adding noise perturbation. We additionally test on alternative ways of fusing $e_s, e_t$. Aligning noise $z$ with $e_s$ before concatenating with $e_t$ (row 4) improves task accuracy and achieves competitive DP, EOpp and EO, but leaks sensitive information as attacker accuracy increases. Fusing task and sensitive embedding with self-attention (row 5) preserves more sensitive information thus slightly hurting the fairness outcomes. Overall, simple concatenation of noisy $e_s$ with $e_t$ achieves the balance between accurate task prediction, fair outcomes and reducing sensitive information leakage.

Table 9: **Effect of noise and projection choices on fairness and utility.** We assess variants of task classification head inputs constructed from frozen backbone output without having any adapter $h$, sensitive embedding $e_s$, task embedding $e_t$, and injected noise $z$. The comparison highlights how noise injection, projection, and attention choices influence task performance and fairness. All reported values are scaled by $\times 10^2$ and show performance from a single run with the same seed. Task: Expression (Smiling); Sensitive attribute: Gender (Male).

| $Task^{clf}$ **Inputs** | **Acc**(↑) | **BAcc**(↑) | **DP**(↓) | **EOpp**(↓) | **EO**(↓) | **Att.Acc**(↓) |
|---|---|---|---|---|---|---|
| $[h]$ | 90.2 | 90.1 | 10.5 | 1.5 | 2.3 | 98.8 |
| $[e_s, e_t]$ | 92.9 | 92.9 | 10.1 | 2.4 | 3.0 | 98.5 |
| $[z, h]$ | 86.4 | 86.5 | 7.0 | 1.4 | 4.1 | 89.5 |
| $[\frac{\langle z^i, e_s^i \rangle}{\|e_s^i\|} e_s^i, e_t]$ | 92.2 | 93.2 | 9.8 | 0.8 | 2.1 | 98.8 |
| Attn $(e_s + z, e_t)$ | 91.9 | 91.7 | 10.2 | 1.3 | 2.8 | 54.5 |
| FairNVT | 93.1 | 92.8 | 9.1 | 0.7 | 2.5 | 52.1 |

**Effect of ablating model components.** Table 10 presents ablation results for different model components. The results are consistent with the main findings in Section 4: the DP loss primarily drives fairness improvements, noise injection reduces sensitive-attribute leakage, and the orthogonality loss further enhances fairness with minimal impact on task performance. Model components also exhibit interacting effects; in particular, combining DP loss with noise injection further decreases DP, EOpp, and EO scores, indicating that enhancing representation-level fairness can align with improvements in prediction-level fairness.

Table 10: **Ablation of FairNVT components on CelebA.** We toggle Demographic Parity loss (DP), Orthogonality loss (Orth), and Noise injection (Noise). ✓and ✗means the component is present and absent respectively. All reported values are scaled by $\times 10^2$ and show performance from a single run with the same seed.

| | Fair Loss | Orth Loss | Noise | **Acc**(↑) | **BAcc** (↑) | **DP** (↓) | **EOpp**(↓) | **EO**(↓) | **Att Acc**(↓) |
|---|---|---|---|---|---|---|---|---|---|
| | ✗ | ✗ | ✗ | 93.1 | 92.9 | 14.1 | 3.8 | 3.8 | 98.9 |
| | ✗ | ✓ | ✓ | 93.2 | 92.8 | 14.5 | 4.8 | 4.8 | 52.9 |
| | ✓ | ✗ | ✓ | 92.8 | 92.7 | 9.8 | 0.2 | 3.1 | 53.0 |
| **Task: Expression (Smiling)** | ✓ | ✓ | ✗ | 93.0 | 93.1 | 9.3 | 0.4 | 3.7 | 99.0 |
| | ✓ | ✗ | ✗ | 92.7 | 92.7 | 8.6 | 0.8 | 4.4 | 98.9 |
| **Sens.: Gender (Male)** | ✗ | ✓ | ✗ | 93.4 | 93.1 | 14.3 | 4.0 | 4.0 | 99.0 |
| | ✗ | ✗ | ✓ | 93.0 | 92.7 | 14.8 | 4.8 | 4.8 | 54.2 |
| | ✓ | ✓ | ✓ | 93.1 | 92.8 | 9.1 | 0.7 | 2.5 | 52.1 |

**Sensitivity of loss weight coefficients.** We analyze the effect of loss weight coefficients in Table 11, using the task and sensitive attributes *expression (smiling)* and *gender (male)*, respectively. A moderate orthogonality loss weight consistently achieves the best balance between task accuracy and fairness metrics, indicating that this setting effectively disentangles task and sensitive embeddings without degrading representation quality. Increasing the DP loss weight improves prediction-level fairness, particularly for demographic parity difference, which it directly optimizes, though with a gradual trade-off in task performance. Because EO and

EOpp condition on specific label groups, they are naturally more sensitive to small prediction variations, yet we observe stable improvements at moderate DP weights. Overall, these trends highlight that the loss weights control the fairness–utility balance in a predictable manner, and tuning them allows FairNVT to adapt robustly across datasets and attribute combinations.

Table 11: **Sensitivity of loss weight coefficients.** We evaluate the performance of FairNVT when the loss weight coefficients changes. All reported values are scaled by $\times 10^2$ and show performance from a single run with the same seed. Task: Expression (Smiling); Sensitive attribute: Gender (Male).

|  | Level | Acc($\uparrow$) | BAcc($\uparrow$) | DP($\downarrow$) | EOpp($\downarrow$) | EO($\downarrow$) | Att.Acc($\downarrow$) |
|---|---|---|---|---|---|---|---|
|  | 0 | 92.8 | 92.7 | 9.9 | 0.2 | 3.1 | 53.0 |
| **Orth Loss** | 0.01 | 92.8 | 92.8 | 10.2 | 0.4 | 2.5 | 53.2 |
|  | 0.1 | 93.1 | 92.8 | 9.1 | 0.7 | 2.5 | 52.1 |
|  | 1.0 | 92.8 | 92.7 | 10.4 | 0.7 | 2.4 | 52.1 |
|  | 0 | 93.2 | 92.8 | 14.5 | 4.8 | 4.8 | 52.9 |
| **DP Loss** | 0.01 | 93.0 | 92.7 | 14.3 | 4.2 | 4.2 | 53.1 |
|  | 0.3 | 93.1 | 92.8 | 9.1 | 0.7 | 2.5 | 52.1 |
|  | 1.0 | 92.1 | 92.3 | 5.7 | 3.5 | 6.1 | 53.4 |

**Sensitivity of embedding clipping.** As discussed in Section 4, we clip the embeddings to an upper bound $C$ before adding noise, which helps control the obfuscation of sensitive information. We analyze the sensitivity of the model to different values of the clipping threshold $C$. Changing $C$ under a fixed noise multiplier ($\sigma$) has a combined effect: it alters the embedding magnitude while also changing the effective noise level, since the noise variance $\sigma^2 C^2$ scales with $C$ (Table 12, rows 1-3). In this setting, smaller $C$ values degrade representation-level fairness, as reflected by higher attacker accuracies. When controlling for noise variance (Table 12, rows 4-6), we observe that varying $C$ produces no significant change in either task accuracy or fairness metrics, suggesting that the clipping operation itself has limited influence once the noise scale is fixed.

Table 12: **Sensitivity of embedding clipping threshold.** We evaluate the performance of FairNVT when the clipping threshold changes. All reported values are scaled by $\times 10^2$ and show performance from a single run with the same seed. Task: Expression (Smiling); Sensitive attribute: Gender (Male).

|  | Level | Acc($\uparrow$) | BAcc($\uparrow$) | DP($\downarrow$) | EOpp($\downarrow$) | EO($\downarrow$) | Att.Acc($\downarrow$) |
|---|---|---|---|---|---|---|---|
| **Clip Threshold** (with same noise multiplier) | 1 | 93.1 | 93.1 | 9.7 | 0.4 | 2.9 | 89.0 |
|  | 10 | 93.1 | 92.8 | 9.1 | 0.7 | 2.5 | 52.1 |
|  | 100 | 92.3 | 92.2 | 10.6 | 0.9 | 1.9 | 53.1 |
| **Clip Threshold** (with same noise amount) | 1 | 92.8 | 92.8 | 9.8 | 0.2 | 2.9 | 52.6 |
|  | 10 | 93.1 | 92.8 | 9.1 | 0.7 | 2.5 | 52.1 |
|  | 100 | 92.8 | 92.8 | 9.7 | 0.3 | 2.7 | 52.2 |

Table 13: **Image-Based Classification task:** Comparing our method with baselines on CelebA (Liu et al., 2015) dataset. All reported values are scaled by $\times 10^2$. We report the mean and standard deviation over 3 runs.

| Method | Acc(↑) | BAcc(↑) | DP(↓) | EOpp(↓) | EO(↓) |
|---|---|---|---|---|---|
| Vanilla | 89.9 (0.1) | 89.4(0.3) | 10.0(0.6) | 2.1(0.3) | 6.1(1.2) |
| ViT-FSCL | 88.7 (0.1) | 88.0(0.1) | 7.4(0.8) | **0.7(0.2)** | 2.5(0.1) |
| FairViT | 92.5(0.2) | 91.9(0.2) | 5.6(0.3) | 1.8(0.9) | 2.3(0.2) |
| FairVPT | 91.9(0.2) | 91.4(0.2) | **1.7(1.0)** | 1.7(1.0) | **2.0(0.3)** |
| FairNet | 92.1(0.3) | 91.6(0.3) | 1.7(2.0) | 1.8(1.0) | 2.1(0.3) |
| FairNVT(Ours) | **92.8(0.1)** | **92.1(0.1)** | 5.8(0.3) | 1.7(1.0) | 2.2(0.2) |

(a) Task: Expression; Sensitive Attribute: Age (Young)

| Method | Acc(↑) | BAcc(↑) | DP(↓) | EOpp(↓) | EO(↓) |
|---|---|---|---|---|---|
| Vanilla | 81.6 (0.2) | 63.2(0.2) | 33.1(2.7) | 40.3(3.0) | 36.8(5.2) |
| ViT-FSCL | 80.4(1.1) | 64.8(0.0) | 24.7(0.2) | 35.2(0.1) | 24.7(0.2) |
| FairViT | 81.9(0.3) | 66.9(0.4) | 20.4(0.5) | 30.6(1.2) | 19.8(0.9) |
| FairVPT | **83.0(0.5)** | 61.1(0.5) | 17.0(0.4) | 25.2(0.9) | 15.7(1.0) |
| FairNet | 81.2(1.0) | 61.2(0.6) | 16.4(0.5) | 18.3(0.8) | 13.0(1.0) |
| FairNVT(Ours) | 81.2(0.1) | **67.4(0.5)** | **8.1(0.6)** | **8.2(1.8)** | **8.3(1.8)** |

(b) Task: Big Nose; Sensitive Attribute: Gender (Male)

| Method | Acc(↑) | BAcc(↑) | DP(↓) | EOpp(↓) | EO(↓) |
|---|---|---|---|---|---|
| Vanilla | 84.4(0.6) | 81.2(0.5) | 10.3(1.0) | 8.5(1.0) | 7.7 (2.0) |
| ViT-FSCL | 83.3 (0.6) | 77.9(2.0) | **5.3(0.8)** | 2.0(0.4) | **1.3(0.3)** |
| FairViT | 86.6(0.4) | 83.7(0.3) | 9.0(0.5) | 3.5(1.0) | 2.8(0.6) |
| FairVPT | 84.2(0.6) | 82.2(0.4) | 7.9(0.6) | 2.8(1.0) | 2.9(0.5) |
| FairNet | 84.4(1.0) | 80.7(1.1) | 8.3(0.7) | **0.5(1.2)** | 1.5(0.8) |
| FairNVT(Ours) | **87.0(0.5)** | **84.0(0.5)** | 7.1(0.7) | 3.4(0.9) | 2.4(0.5) |

(c) Task: Wavy hair; Sensitive Attribute: Age(Young)

| Method | Acc(↑) | BAcc(↑) | DP(↓) | EOpp(↓) | EO(↓) |
|---|---|---|---|---|---|
| Vanilla | 99.1 (0.1) | 94.1(0.3) | 10.8(0.1) | 5.3(1.5) | 4.3 (1.3) |
| ViT-FSCL | 99.1(0.1) | 95.0(0.2) | 10.8(0.4) | 3.9(0.4) | 2.4(0.2) |
| FairViT | 99.0(0.2) | 98.0(0.2) | 10.0(0.3) | 0.8(0.3) | **0.6(0.3)** |
| FairVPT | 99.4(0.1) | 97.0(0.2) | **9.6(0.3)** | 2.1(0.3) | 1.2(0.4) |
| FairNet | 99.3(0.1) | 93.9(0.9) | 10.4(0.6) | **0.3(1.1)** | 1.2(0.3) |
| FairNVT(Ours) | **99.6(0.0)** | **98.7(0.0)** | 11.2(0.1) | 0.7(0.4) | 0.7(0.4) |

(d) Task: Wearing glasses; Sensitive Attribute: Gender (Male)

| Method | Acc(↑) | BAcc(↑) | DP(↓) | EOpp(↓) | EO(↓) |
|---|---|---|---|---|---|
| Vanilla | 99.1 (0.1) | 95.4(0.2) | 13.7(0.2) | 7.0(0.6) | 6.3(1.6) |
| ViT-FSCL | 99.0(0.1) | 95.1(1.1) | 13.1(0.5) | 5.9(0.1) | 3.3(0.2) |
| FairViT | 99.1(0.3) | 97.4(0.4) | 13.0(0.7) | 2.7(0.6) | 2.9(0.5) |
| FairVPT | 99.4(0.1) | 96.8(0.3) | 12.9(0.7) | 1.2(1.0) | 2.4(0.7) |
| FairNet | **99.6(0.3)** | **98.5(0.4)** | 13.5(0.4) | **1.0(1.2)** | **0.7(1.3)** |
| FairNVT(Ours) | 99.3(0.2) | 96.9(0.5) | **12.4(1.0)** | 3.0(1.1) | 2.8(1.3) |

(e) Task: Wearing Glasses; Sensitive Attribute: Age (Young)

| Method | Acc(↑) | BAcc(↑) | DP(↓) | EOpp(↓) | EO(↓) |
|---|---|---|---|---|---|
| Vanilla | 85.5(0.3) | 85.2(0.4) | 9.6(0.4) | 4.7(0.6) | 4.6(0.9) |
| ViT-FSCL | 82.6(0.5) | 81.8(0.6) | 7.5(2.0) | 1.3(0.7) | 2.5(1.7) |
| FairViT | 93.4(0.1) | 93.3(0.2) | 9.0(0.4) | 1.6(0.3) | 1.5(0.4) |
| FairVPT | 92.7(0.1) | 92.7(0.1) | 9.3(0.3) | **0.3(0.8)** | **0.6(0.7)** |
| FairNet | 91.0(0.5) | 90.9(0.4) | 8.1(0.4) | 1.1(0.7) | 1.3(0.8) |
| FairNVT(Ours) | **93.7(0.1)** | **93.7(0.1)** | **6.3(0.8)** | 0.9(0.1) | 1.5(0.6) |

(f) Task: Mouth Slightly Open; Sensitive Attribute: Gender(Male)

| Method | Acc(↑) | BAcc(↑) | DP(↓) | EOpp(↓) | EO(↓) |
|---|---|---|---|---|---|
| Vanilla | 84.7(1.8) | 83.9(1.7) | 7.4(1.5) | 1.8(1.3) | 6.1(1.8) |
| ViT-FSCL | 83.8(0.1) | 82.9(0.1) | 5.7(1.2) | 1.5(1.3) | 2.3(1.1) |
| FairViT | 93.4(0.3) | 93.1(0.2) | 7.0(0.3) | 0.8(0.2) | 0.9(0.3) |
| FairVPT | 92.0(0.1) | 91.8(0.1) | **4.4(0.5)** | 1.8(1.0) | 1.0(0.4) |
| FairNet | 93.3(0.3) | 93.1(0.4) | **0.9(0.5)** | 0.7(0.9) | 1.0(0.5) |
| FairNVT(Ours) | **94.0(0.0)** | 93.7(0.2) | 4.6(0.2) | **0.3(0.1)** | **0.6(0.1)** |

(g) Task: Mouth Slightly Open; Sensitive Attribute: Age(Young)

Table 14: **Qualitative BIOS examples.** We show pairs of biography snippets that differ only in gender indicators. For each snippet, we display the model's predicted occupation and the prediction score for the ground-truth label. Vanilla model predictions vary substantially across genders, suggesting reliance on gender cues, whereas FairNVT yields more stable scores and consistent predictions, indicating improved robustness to gender indicators.

| ID | BIO Snippet | Vanilla | FairNVT |
|:---:|---|:---:|:---:|
| 1 | He specializes in development economics, household economics, and personnel economics. In 2003 he received his Ph.D. in Economics from the London School of Economics... | professor (0.903) | professor (0.992) |
| 1 | She specializes in development economics, household economics, and personnel economics. In 2003 she received her Ph.D. in Economics from the London School of Economics... | professor (0.882) | professor (0.993) |
| 2 | Prosper was born and raised in Miami Beach, FL. He received his Bachelor's degree from Emory University and graduated with honors from the University of Miami School of Law... | attorney (0.971) | attorney (0.971) |
| 2 | Prosper was born and raised in Miami Beach, FL. She received her Bachelor's degree from Emory University and graduated with honors from the University of Miami School of Law... | attorney (0.939) | attorney (0.970) |
| 3 | She has been travelling the world, and worked, amongst others, on a documentary photography project in India with an orphanage... | photographer (0.643) | photographer (0.908) |
| 3 | He has been travelling the world, and worked, amongst others, on a documentary photography project in India with an orphanage... | photographer (0.729) | photographer (0.864) |
| 4 | She studied at EFET Paris and NYU New-York respectively. While working in a post-production, she develops her own photographic concept... | photographer (0.664) | photographer (0.966) |
| 4 | He studied at EFET Paris and NYU New-York respectively. While working in a post-production, he develops his own photographic concept... | photographer (0.804) | photographer (0.941) |
| 5 | He attended the University of California, San Francisco (UCSF), School of Medicine and subsequently trained at Children's Hospital Los Angeles for residency... | physician (0.717) | physician (0.818) |
| 5 | She attended the University of California, San Francisco (UCSF), School of Medicine and subsequently trained at Children's Hospital Los Angeles for residency... | physician (0.826) | physician (0.755) |

| ID | BIO Snippet | Vanilla | FairNVT |
|---|---|---|---|
| 6 | Dr. Cottrell attended medical school at the University of Missouri-Columbia School of Medicine. He is in-network for Anthem, Blue Cross/Blue Shield, Blue Shield, and more. | physician (0.773) | physician (0.923) |
| 6 | Dr. Cottrell attended medical school at the University of Missouri-Columbia School of Medicine. She is in-network for Anthem, Blue Cross/Blue Shield, Blue Shield, and more. | physician (0.791) | physician (0.920) |
| 7 | After spending two years at the University of Iowa, Kyle transferred to Chapman University, where he directed a superhero noir titled The League, about the 1960's superhero labor union of Chicago... | filmmaker (0.944) | filmmaker (0.808) |
| 7 | After spending two years at the University of Iowa, Kylie transferred to Chapman University, where she directed a superhero noir titled The League, about the 1960's superhero labor union of Chicago... | filmmaker (0.925) | filmmaker (0.866) |
| 8 | She has been a successful Dentist for the last 16 years. She is a BDS. She is currently associated with SMII Dental Art Studio in Koregaon Park, Pune... | dentist (0.991) | dentist (0.997) |
| 8 | He has been a successful Dentist for the last 16 years. He is a BDS. He is currently associated with SMII Dental Art Studio in Koregaon Park, Pune... | dentist (0.966) | dentist (0.943) |
| 9 | Downs was a fellow at Northwestern University's Academy for Alternative Journalism in 2004, and he earned a degree in English literature from University of California at Santa Barbara in 2002... | journalist (0.437) | journalist (0.917) |
| 9 | Downs was a fellow at Northwestern University's Academy for Alternative Journalism in 2004, and she earned a degree in English literature from University of California at Santa Barbara in 2002... | journalist (0.480) | journalist (0.885) |
| 10 | She graduated with honors in 2012. Having more than 5 years of diverse experiences, especially in NURSE PRACTITIONER, Melissa R Kludt affiliates with many hospitals including... | nurse (0.917) | nurse (0.890) |
| 10 | He graduated with honors in 2012. Having more than 5 years of diverse experiences, especially in NURSE PRACTITIONER, Miles R Kludt affiliates with many hospitals including... | nurse (0.624) | nurse (0.926) |

