# OpenReview forum: "FairNVT: Fair Classification via Noise Injection in Vision Transformers"
_TMLR — Under review for TMLR_

### Review · Reviewer_pKZb · 2026-06-10

**Summary Of Contributions:**

The paper proposes FairNVT, a parameter-efficient debiasing method for frozen transformer encoders. The method trains separate task and sensitive adapters, injects calibrated Gaussian noise into the sensitive embedding, concatenates the noised sensitive branch with the task embedding, and optimizes task, sensitive-attribute, orthogonality, and demographic-parity losses. Experiments are conducted mainly on CelebA and UTKFace with ViT-B/16, with an additional BIOS/BERT experiment in the appendix. The reported results show improved DP/EO/EOpp gaps and substantially lower sensitive-attribute attacker accuracy while maintaining competitive target-task accuracy.

Strengths
- The goal of jointly improving representation-level and prediction-level fairness for pretrained encoders is important. This matters because many practical systems adapt large frozen backbones rather than training fair models from scratch.
- The proposed method is simple and parameter-efficient. The split-adapter design and noise injection mechanism are easy to implement on top of frozen ViT/BERT-style encoders.
- The paper includes several useful ablations, including the fairness loss, orthogonality loss, noise strength, clipping threshold, attacker depth, and inference-time noise draws. These analyses help isolate which parts of the method contribute to the reported fairness gains.

Weaknesses
- The theoretical argument is overstated. Marginal independence between the learned representation and sensitive attribute supports demographic parity for predictors using that representation, but it does not imply equalized odds or equal opportunity without an additional conditional-independence assumption given the target label.
- The representation-fairness evidence is incomplete. Attacker accuracy is mainly reported on the stochastic fused embedding, so the results may reflect noise-based obfuscation rather than removal of sensitive information from reusable deterministic representations such as the task adapter embedding.
- The baseline and model-capacity controls are not fully convincing. FairNVT uses sensitive supervision, an additional sensitive adapter, and a DP loss, but the comparisons are not presented as matched fairness-utility Pareto curves under a common validation criterion.
- The treatment of uncertainty is useful but still limited. The paper reports three-run standard deviations, but it is not always clear whether these include training randomness, inference-time noise randomness, or both, and the stochastic inference mechanism makes this distinction important.

**Additional Comments:**

NA

**Audience:**

Yes

**Audience Explanation:**

TMLR readers working on fairness, representation learning, and parameter-efficient adaptation would likely find the idea of targeted noise injection in a learned sensitive subspace interesting. The vision and text experiments are also relevant to practical debiasing of pretrained encoders. The paper would be substantially more useful if it more carefully established what sensitive information is actually removed from the learned representation versus what is hidden by inference-time randomness.

**Broader Impact Concerns:**

The paper uses face datasets with perceived gender and age labels, including binary groupings. This is standard in the benchmark literature but ethically sensitive. The broader-impact discussion should explicitly note that such labels are socially constructed, noisy, and potentially harmful if used outside controlled auditing research. The authors should also clarify that lower benchmark disparity does not certify fairness in real deployments, especially under distribution shift or for unmodeled intersectional groups.

**Claims And Evidence:**

No

**Claims Explanation:**

The empirical results are promising, but several central claims need tighter support. First, Appendix B claims that Z independent of S implies DP, EO, and EOpp. The DP part is valid for predictors depending only on Z, but EO/EOpp require independence after conditioning on Y. This does not follow from marginal independence and is especially important when Y and S are correlated, as in the face-attribute tasks. The paper should either correct the statement or add the needed assumptions and discuss when they are plausible.

Second, the attacker evaluation may conflate true sensitive-information removal with stochastic obfuscation. Since the task classifier receives [e_t, e_s + noise], an attacker on the fused embedding can be degraded by high-variance noise even if e_t still contains recoverable sensitive information. This matters because e_t is deterministic and could be reused downstream. The paper should report attacker accuracy and balanced attacker accuracy on e_t alone, clean e_s, noised e_s, and the fused embedding. It should also evaluate attackers with access to multiple independent noisy samples for the same input, or to the averaged fused embedding.

Third, the comparison protocol should better separate debiasing from capacity, tuning, and objective choices. The main text says hyperparameters are tuned for highest task accuracy, which is not a neutral criterion for comparing fairness methods. A stronger evaluation would report validation-selected fairness-utility Pareto curves for FairNVT and baselines at matched accuracy or balanced-accuracy bands. Important controls include a split-adapter + DP-loss baseline without noise and a task-adapter-only + DP-loss baseline with the same trainable parameter budget.

Finally, the use of statistics and treatment of uncertainty are only partially appropriate. Reporting three-run standard deviations is helpful, but the paper should clarify whether the reported uncertainty comes from training seeds, inference-time noise draws, or both. Because FairNVT is stochastic at inference time, expected metrics over repeated noise draws and uncertainty over those draws are part of the method's behavior, not merely implementation details.

**Requested Changes:**

- Correct the theoretical claim connecting representation fairness to EO/EOpp. State that Z independent of S supports DP for predictors using Z, but EO/EOpp require additional assumptions such as Z independent of S conditional on Y, or provide a valid bound for EO/EOpp in terms of conditional distribution distances.
- Strengthen the representation-leakage evaluation. Add probes on e_t, clean e_s, noised e_s, and [e_t, noised e_s]; include balanced attacker accuracy for all sensitive attributes; and test an attacker that receives multiple noise draws per input or the mean over repeated draws.
- Add matched fairness-utility comparisons. For CelebA and UTKFace, compare FairNVT against baselines at similar accuracy/BAcc levels using validation-selected trade-off curves, and include controlled variants such as task adapter + DP loss, split adapters + DP loss, and split adapters + orthogonality + DP loss without noise.
- Report sensitivity to the sensitive-label imbalance and to batch composition in the DP loss, especially for age on CelebA and UTKFace.
- Clarify whether the reported three-run standard deviations include inference-time noise randomness, training randomness, or both; report expected metrics and uncertainty over several independent inference noise draws.
- Expand the BIOS experiment beyond DP and attacker accuracy, or explain why only DP is reported; for multi-class occupation, macro group gaps or per-class worst-case gaps would be more informative.
- Discuss deployment assumptions: FairNVT requires sensitive labels during training, uses perceived gender/age annotations, and makes stochastic predictions at inference. These are important practical limitations.

---

> ### Author Response · Authors · 2026-07-14
>
> We thank the reviewer for the constructive feedback, which has helped us clarify both the motivation and empirical evaluation of FairNVT.
>
> A central clarification is that FairNVT is designed to reduce, rather than eliminate, sensitive information in the learned representation. While statistical independence between the representation and sensitive attribute motivates the framework design, we do not claim to achieve complete disentanglement in practice. To better reflect this distinction, we removed attacker accuracy on the embedding prior to the downstream classifier as a primary comparison metric with baselines and instead present it in ablation studies to evaluate the behavior of our proposed framework. We refined Section 2 to clarify the term representation-level fairness to its practical meaning of ``reducing sensitive attribute leakage,'' revised the mathematical discussion in Appendix B to present the design as an intuitive motivation rather than a guarantee, and updated the experiment and ablation discussions in Section 4.
>
> The requested changes are summarized below. The changes made to the paper are highlighted in red.
>
> **``Correct the theoretical claim connecting representation fairness to EO/EOpp.''**
>
> We revised Appendix B to explicitly distinguish the assumptions required for demographic parity and equalized odds/equal opportunity. We stated the additional conditional independence assumption needed for EO/EOpp and clarified that the discussion provides intuition for the framework design rather than a formal guarantee.
>
> **``Strengthen the representation-leakage evaluation.''**
>
> We added Table 3 in Section 4.2, which evaluates attacker accuracy and balanced accuracy on the task embedding, sensitive embedding, noised sensitive embedding, and fused embedding. We further evaluate stronger attackers and the effect of averaging multiple independent noise realizations. These results are discussed in the paragraph ``Evaluate sensitive attribute leakage with attacker.''
>
> **``Add matched fairness--utility comparisons.''**
>
> We added Pareto curve comparisons with all baselines in Figure 2, Section 4.1, and controlled ablation variants in Figure 4, Section 4.2, together with corresponding discussions. We also keep the comparison based on the highest task-accuracy model for each method, as this reflects the common deployment scenario of selecting a single model, while the Pareto curves provide a complementary view of the overall fairness--utility trade-off.
>
> **``Report sensitivity to the sensitive-label imbalance and batch composition.''**
>
> We added sensitive-label imbalance numbers in the ``Datasets and tasks'' paragraph in Section 4. Notably, the Age attribute in CelebA is the most imbalanced, with the majority/minority groups having 77\% and 23\% of the samples, respectively. The per-batch imbalance is quite similar to the overall distribution; for example, for the Age attribute in CelebA, the first three batches are distributed as 75\%/25\%, 80\%/20\%, and 79\%/21\%. We add an evaluation of model performance with smaller batch sizes of 128 and 64 in Appendix E Table 6. We acknowledge that the current fairness loss does not particularly handle severe class imbalance in the sensitive attribute, as the minority-class estimate might suffer from high variance. We note this limitation and leave it for future work.
>
> **``Clarify the source of randomness when reporting standard deviations.''**
>
> We added clarification in Section 4.1, in the paragraph ``Comparison at the best task accuracy.'' For the baseline methods, the reported standard deviations reflect randomness from model initialization and training. For FairNVT, they additionally include the stochasticity introduced by Gaussian noise at inference, thereby capturing both training- and inference-time sources of randomness, i.e. we evaluate the full pipeline for 3 times. We add an evaluation in Appendix E Table 7 on model performance when fixing the trained model parameters and make inference 3 times where randomness come from the random noise at inference time only. We observe that the variations in performance is small.
>
> **``Expand the BIOS experiment beyond DP.''**
>
> We report DP for the multi-class task attribute case only because equalized odds and equal opportunity condition on the true task label and do not have a unified multi-class definition, whereas demographic parity difference does not condition on the true task label and is therefore directly applicable to multi-class cases. We modified the footnote on page 20 to explain this when discussing the BIOS results in Appendix D.2.
>
> **``Discuss deployment assumptions.''**
> We added a discussion of the practical limitations of FairNVT in the conclusion, and added a broader impact concern section after conclusion.

---

### Review · Reviewer_7BwZ · 2026-06-19

**Summary Of Contributions:**

The paper proposes patching a pretrained transformer to enhance its fairness by changing its representation through a learned noised head. The intervention consists first in trying to separate sensitive infromation and task relevant embeddings. Both task and sensitive information are classified by losses such that the network can learn the proper sensitive and task embeddings. As the network learns these embeddings, noise is injected from the sensitive embeddings to the task embeddings to try to further disentangle task embeddings from sensitive information.

The whole architecture, objective as well as the background are well exposed. Experiments on vision tasks with ViTs demonstrate that the proposed method improves most fairness metrics while preserving and even sometimes improving the vanilla architecture. Qualitative analysis shows that the proposed method pay more attention to relevant images details. Ablations show how each component contributes to the overall performance. The appendix contains more results on text as well as more ablations and qualitative results.

**Audience:**

Yes

**Audience Explanation:**

I'm no specialist in fairness but I suppose that the community will appreciate this new approach. Most importantly it appears to be a well thought and rigorously conducted experimental project which, even if it may not become the state of the art, provides a great template to make similar contributions.

**Broader Impact Concerns:**

None foreseen

**Claims And Evidence:**

Yes

**Claims Explanation:**

The experiments are done on several datasets used in the literature, both in vision and in text (in the appendix). Many ablations are done to fully grasp the benefits of each component. The setup is thoroughly detailed with both clear architecture graphs and mathematical formulations. Experimental details are given in Appendix C and appear reproducible.

**Requested Changes:**

- Why is the noise injected sensitive embedding simply concatenated to the token embedding? A priori the whole architecture could then simply ignore the whole part corresponding do the noise embedding. If the architecture works well it should progressively only pay attention to the token embedding part (so just the first part), and in that case the whole noise injection is pointless (one could just remove it from the start). One could also have tried reinjecting it by adding the noised sensitive embeddings to the token embeddings.
- In Figure 1, would it be possible to somehow add how those modules are integrated throughout the depth of the model? The figure makes the reader think these adapters are just done at the end while the fact that they are at every layer is an important feature.
- Are the models compared with equal number of parameters? It seems that the method may add parameters which could explain why it performs better than the vanilla baseline.

---

> ### Author Response · Authors · 2026-07-14
>
> We thank the reviewer for the insightful comment. We address the point below.
>
> **``Why is the noise-injected sensitive embedding simply concatenated to the token embedding?''**
>
> We understand the reviewer's concern that the downstream classifier could ignore the noised sensitive branch by assigning it a negligible weight. However, this would only be optimal and practically doable if the sensitive and task information were perfectly separated already. In our setting, both adapters are initialized from the same frozen representation, where task and sensitive information are entangled and initially $e_t \approx e_s \approx z$. Consequently, the sensitive branch initially also contains task-relevant information, and simply ignoring it would degrade task performance. Instead, the split-adapter architecture together with the orthogonality objective encourages task and sensitive information to gradually separate into different channels during training. Noise injection then makes the sensitive branch less reliable for prediction, encouraging the classifier to rely more on the task branch, while the fairness loss discourages reliance on the residual sensitive information that remains in the fused representation. We have revised Appendix~B to better explain this design intuition by separating: (1) how statistical independence relates to prediction-level fairness, and (2) how this intuition motivates the proposed architecture for debiasing frozen representations.
>
> We also added new ablation studies to support this design choice. Table 3 in Section 4.2 evaluates sensitive attribute leakage on the task, sensitive, noised sensitive, and fused embeddings, demonstrating the intended effect of each component. In addition, we evaluated alternative fusion strategies in Appendix D, Table 7, Row 5, including an attention-based fusion between $e_t$ and $e_s^{\mathrm{noised}}$, which is conceptually similar to adding the embeddings, and this alternative resulted in lower task performance than simple concatenation.
>
> The other requested changes are summarized below. The changes made to the paper are highlighted in red.
>
> **`In Figure 1, ... add how those modules are integrated throughout the depth of the model.''**
>
> We modified the plots with a double line arrow indicating that the Adapter attaches trainable parameters to each Transformer layers. We also modified the footnote in page 4 hoping to better guide readers to the Adapter architecture plot in its documentation file.
>
> **``Are the models compared with an equal number of parameters?''**
>
> We list the hyperparameter search ranges for all methods in the ``Implementation Details'' paragraph in Appendix C. For each method, we tune the key hyperparameters that most strongly affect task and fairness performance using grid search. Although different methods have different numbers of tunable hyperparameters, the search effort is comparable across methods, and each baseline is evaluated using its best validation-selected configuration.
>
> Based on other reviewers' feedback, we have also refined the experimental evaluations. The additions include Pareto curve comparisons of fairness--utility trade-offs in Figures 2 and 4, comparisons with an additional baseline in Table 1 and Table 12 in Appendix D, and new ablation studies on sensitive attribute leakage across different learned representations in Table 3. We summarize these changes here for the reviewer's convenience.

---

### Review · Reviewer_kMde · 2026-06-30

**Summary Of Contributions:**

The paper proposes FairNVT, a lightweight framework for improving both representation-level and prediction-level fairness in pretrained transformer encoders. It freezes the backbone, learns separate task and sensitive-attribute adapters, injects calibrated Gaussian noise into the sensitive embedding, and combines classification, orthogonality, and demographic-parity losses to reduce sensitive-attribute leakage while preserving task performance. Experiments on CelebA, UTKFace, and an additional BIOS text setting show improved fairness metrics and substantially lower attacker accuracy.

The main strengths are the simple and parameter-efficient design, the unified treatment of representation and prediction fairness, and empirical reductions in sensitive information leakage. The main weaknesses are the reliance on sensitive-attribute labels during training, the relatively narrow experimental scope (datasets and baselines), and the limited evidence for representation fairness beyond attacker accuracy and noise-sensitivity analyses.

**Audience:**

Yes

**Audience Explanation:**

The topic is relevant to several communities, including fairness in machine learning, representation learning. The paper’s attempt to connect representation-level fairness with prediction-level fairness through a lightweight adapter-and-noise-injection mechanism is practical, because it keeps the pretrained backbone frozen and is compatible with a wide range of pretrained transformer encoders.

The reported reductions in sensitive-attribute leakage and improvements in DP/EOpp/EO would be useful to researchers working on fair representation learning, debiasing pretrained models, and efficient fairness interventions. However, the findings are not necessarily of broad interest to all TMLR readers, since the main empirical evidence is concentrated on relatively standard fairness benchmarks and binary classification settings.

**Broader Impact Concerns:**

The method requires sensitive-attribute labels during training, such as perceived gender or age in facial datasets. This raises concerns about data collection, annotation quality, and the risk of reinforcing socially constructed or noisy demographic categories. The paper notes that these labels are used for modeling and fairness evaluation, but the broader implications of relying on such annotations deserve more discussion.

**Claims And Evidence:**

Yes

**Claims Explanation:**

The core claims are supported by relevant and clearly presented evidence. The experiments evaluate both task utility and fairness using Acc, DP, EOpp, EO, and attacker accuracy, and the results on CelebA and UTKFace generally show improved fairness with limited loss of task performance. The ablations also support the importance of noise injection, fairness loss, and orthogonality regularization.

However, the evidence is not fully conclusive. Representation-level fairness is mainly measured through MLP attacker accuracy, which does not prove that sensitive information is completely removed. The main experiments are also concentrated on binary facial-attribute classification, so broader claims about general applicability to transformer encoders are somewhat under-supported. Overall, the evidence supports the paper’s main benchmark-level claims, but not its strongest generalization claims.

**Requested Changes:**

1. Broaden the baseline comparison.

The current baseline set is somewhat limited for supporting the paper’s claims about dual-level fairness. The authors should include stronger and more recent fair representation / fair adaptation baselines, especially methods that also intervene on internal representations. For example, FairNet [1] appears to be a closely related baseline and should be added if applicable. This would make the empirical comparison more convincing.

2. Expand beyond standard classification tasks.

The main experiments are concentrated on binary facial-attribute classification, with text classification only in the appendix. To support broader claims about applicability to pretrained transformer encoders, the paper should include more diverse task settings, such as multi-class classificatio, or more complex fair vision-language / NLP tasks.

3. Report fairness–utility Pareto curves.

Since the method depends on noise strength and fairness-loss weights, single-point comparisons may not fully characterize the trade-off between accuracy and fairness. The authors should report Pareto curves over different hyperparameter settings, comparing FairNVT and baselines under matched accuracy or matched fairness levels.

4. Strengthen the evaluation of representation-level fairness.

Attacker accuracy is useful but not sufficient to establish that sensitive information is removed from the representation. The authors should evaluate stronger probes, different attacker architectures.

5. Improve theoretical justification.

The paper would benefit from a clearer argument connecting calibrated noise injection, reduced sensitive-attribute recoverability, and improved prediction-level fairness.

[1] FairNet: Dynamic Fairness Correction without Performance Loss via Contrastive Conditional LoRA

---

> ### Author Response · Authors · 2026-07-14
>
> We thank the reviewer for the constructive feedback that helped us improve the presentation and evaluation of the proposed method.
>
> We would first like to clarify two central points.
>
> First, FairNVT is designed to reduce, rather than eliminate, sensitive information in the learned representation. While statistical independence between the representation and sensitive attribute motivates the framework design, we do not claim to achieve complete disentanglement in practice. To better reflect this distinction, we removed attacker accuracy on the embedding prior to the downstream classifier as a primary comparison metric with baselines and instead present it in ablation studies to evaluate the behavior of our proposed framework. We refined Section 2 to clarify the term representation-level fairness to its practical meaning of ``reducing sensitive attribute leakage,'' revised the mathematical discussion in Appendix B to present the design as an intuitive motivation rather than a guarantee, and updated the experiment and ablation discussions in Section 4.
>
> Second, we acknowledge that FairNVT requires sensitive attribute labels during training and discuss this as a practical deployment limitation in the conclusion. We note, however, that this assumption is shared by the supervised fairness baselines considered in our experiments. For example, FSCL uses sensitive labels to construct fairness-aware contrastive pairs, FairViT uses them for mask optimization, FairVPT incorporates them during prompt tuning, and FairNet uses them for adversarial fairness learning.
>
> The requested changes are summarized below. The changes made to the paper are highlighted in red.
>
> **``Broaden the baseline comparison.''**
>
> We added FairNet as an additional baseline and updated all relevant comparisons, including Figures 2 and 4, Table 1, and Table 12 in Appendix D, together with the corresponding discussions.
>
> **`Expand beyond standard classification tasks.''**
>
> We note that BIOS is a multi-class text classification benchmark and is included in our evaluation in Appendix D. Similar to FSCL, FairViT, FairVPT, and FairNet, we focus the main experiments on binary classification because standard fairness metrics such as demographic parity, equal opportunity, and equalized odds are naturally defined in this setting. We discuss this limitation in the conclusion and identify broader fairness tasks as future work.
>
> **``Report fairness--utility Pareto curves.''**
>
> We added Pareto curve comparisons with all baselines in Figure 2, Section 4.1, and controlled ablation variants in Figure 4, Section 4.2, together with corresponding discussions. We also keep the comparison based on the highest task-accuracy model for each method, as this reflects the common deployment scenario of selecting a single model, while the Pareto curves provide a complementary view of the overall fairness--utility trade-off.
>
> **``Strengthen the evaluation of representation-level fairness.''**
>
> We added Table 3 in Section 4.2, which evaluates attacker accuracy and balanced accuracy on the task embedding, sensitive embedding, noised sensitive embedding, and fused embedding. We further evaluate stronger attackers and the effect of averaging multiple independent noise realizations. These results are discussed in the paragraph ``Evaluate sensitive attribute leakage with attacker.''
>
> **``Improve theoretical justification.''**
>
> We revised Appendix B to better explain the mathematical intuition underlying the design of the proposed method. We separate the discussion into: (1) how statistical independence relates to prediction-level fairness, and (2) how this intuition motivates the proposed architecture for debiasing frozen representations. We also corrected the assumptions required for demographic parity and equalized odds/equal opportunity, and stated the additional conditional independence assumption required for EO/EOpp, as also noted by Reviewer pKZb.